# Synaptic FUS accumulation triggers early misregulation of synaptic RNAs in a mouse model of ALS

Sonu Sahadevan[1,5], Katharina M. Hembach [1,2,5], Elena Tantardini[1], Manuela Pérez-Berlanga [1], Marian Hruska-Plochan [1], Salim Megat[3], Julien Weber[1], Petra Schwarz[4], Luc Dupuis [3], Mark D. Robinson[2], Pierre De Rossi [1] & Magdalini Polymenidou [1✉]

Mutations disrupting the nuclear localization of the RNA-binding protein FUS characterize a subset of amyotrophic lateral sclerosis patients (ALS-FUS). FUS regulates nuclear RNAs, but its role at the synapse is poorly understood. Using super-resolution imaging we determined that the localization of FUS within synapses occurs predominantly near the vesicle reserve pool of presynaptic sites. Using CLIP-seq on synaptoneurosomes, we identified synaptic FUS RNA targets, encoding proteins associated with synapse organization and plasticity. Significant increase of synaptic FUS during early disease in a mouse model of ALS was accompanied by alterations in density and size of GABAergic synapses. mRNAs abnormally accumulated at the synapses of 6-month-old ALS-FUS mice were enriched for FUS targets and correlated with those depicting increased short-term mRNA stability via binding primarily on multiple exonic sites. Our study indicates that synaptic FUS accumulation in early disease leads to synaptic impairment, potentially representing an initial trigger of neurodegeneration.

[1] Department of Quantitative Biomedicine, University of Zurich, Zürich, Switzerland. [2] Department of Molecular Life Sciences and SIB Swiss Institute of Bioinformatics, University of Zurich, Zürich, Switzerland. [3] Inserm, University of Strasbourg, Strasbourg, France. [4] Institute of Neuropathology, University Hospital Zurich, Zürich, Switzerland. [5] These authors contributed equally: Sonu Sahadevan, Katharina M. Hembach. ✉email: magdalini.polymenidou@uzh.ch

Fused in sarcoma (FUS) is a nucleic acid binding protein involved in several processes of RNA metabolism[1]. Physiologically, FUS is predominantly localized to the nucleus[2] via active transport by transportin (TNPO)[3] and it can shuttle to the cytoplasm by passive diffusion[4,5]. In amyotrophic lateral sclerosis (ALS) and frontotemporal dementia (FTD), FUS mislocalizes to the cytoplasm where it forms insoluble aggregates[6–8]. In ALS, cytoplasmic mislocalization of FUS is associated with mutations that are mainly clustered in the nuclear localization signal at the C-terminal site of the protein[9] and lead to mislocalization of the protein to the cytosol. However, in sporadic FTD, FUS mislocalization occurs in the absence of mutations[10]. FUS is incorporated in cytoplasmic stress granules[5,11] and undergoes concentration-dependent, liquid–liquid phase separation[12,13], which is modulated by binding of TNPO and arginine methylation of FUS[14–17]. This likely contributes to the role of FUS in forming specific identities of ribonucleoprotein (RNP) granules[18,19] and in transporting RNA cargos[20], which is essential for local translation in neurons[21].

Despite the central role of FUS in neurodegenerative diseases, little is known about its function in specialized neuronal compartments, such as synapses. FUS was shown to mediate RNA transport[20] and is involved in stabilization of RNAs that encode proteins with important synaptic functions[22], such as *GluA1* and *SynGAP1*[23,24]. While the presence of FUS in synaptic compartments has been confirmed, its exact subsynaptic localization is debated. Diverging results described the presence of FUS at the pre-synapses in close proximity to synaptic vesicles[25–27], but also in dendritic spines[20] and in association with the postsynaptic density[28]. Confirming a functional role of FUS at the synaptic sites, behavioral and synaptic morphological changes have been observed upon depletion of FUS in mouse models[23,29,30]. Notably, mouse models associated with mislocalization of FUS exhibited reduced axonal translation contributing to synaptic impairments[31]. Synaptic dysfunction has been suggested to be the early event of several neurodegenerative disorders including ALS and FTD[32–36]. The disruption of RNA-binding proteins (RBPs) and RNA regulation could be a central cause of synaptic defects in these disorders.

Previous studies identified nuclear RNA targets of FUS with different cross-linking immunoprecipitation and high-throughput sequencing (CLIP-seq) approaches[22,37–41]. Collectively, these findings showed that FUS binds mainly introns, without a strong sequence specificity, but a preference for either GU-rich regions[22,38,40,41], which is mediated via its zinc finger (ZnF) domain, or a stem-loop RNA[37] via its RNA recognition motif[42,43]. FUS often binds close to alternatively spliced exons, highlighting its role in splicing regulation[22,38,39]. CLIP-seq studies also identified RNAs bound by FUS at their 3′ untranslated regions (3′UTRs) and exons[22,39,41], suggesting a direct role of FUS in RNA transport and regulating synaptic mRNA stability[23,24] and polyadenylation[40]. However, a precise list of synaptic RNAs directly regulated by FUS is yet to be identified.

In this study, we focused on understanding the role of synaptic FUS in RNA homeostasis and the consequences of ALS-causing mutations in FUS on synaptic maintenance. Using superresolution imaging, we confirmed the presence of FUS at both pre and postsynaptic sites of excitatory and inhibitory synapses. Synaptoneurosome preparations from adult mouse cortex, coupled with CLIP-seq uncovered specific synaptic RNA targets of FUS, which encode proteins associated with both glutamatergic and GABAergic networks. In a heterozygous FUS knock-in mouse model, which harbors a deletion in the NLS of FUS allele, thereby mimicking the majority of ALS-causing FUS mutations[44], we found significant increase of synaptic FUS localization. Elevation in synaptic FUS was accompanied by mild and transient synaptic changes. However, RNA-seq analysis revealed age-dependent alterations of synaptic RNA composition including glutamatergic and GABAergic synapses. mRNAs abnormally accumulated at the synapses of 6-month-old ALS-FUS mice were enriched for FUS targets, suggesting that they might be directly regulated by FUS binding. We determined mRNAs with significantly increased stability in ALS-FUS neurons and found them correlated with those abnormally accumulated by 6 months of age in these mice. FUS binding on multiple exonic sites was enriched within aberrantly increased mRNAs pointing to a molecular code for FUS-mediated synaptic RNA regulation. Our data indicate that early synaptic alterations in the GABAergic network precede motor impairments in these ALS-FUS mice[44], and may trigger early behavioral dysfunctions, such as hyperactivity and social disinhibition[45]. Importantly, we show that increased synaptic localization of FUS, in the absence of aggregation, suffices to cause synaptic impairment.

## Results

**FUS is enriched at the presynaptic compartment of mature cortical and hippocampal neurons.** While FUS has been shown at synaptic sites, its exact subsynaptic localization is debated. Some studies described a presynaptic enrichment of FUS in cortical neurons and motoneurons[25,27], whereas others have shown an association of FUS with postsynaptic density (PSD) sites[20,28]. To clarify the precise localization of FUS at the synapses, we first performed confocal analysis in mouse cortex (Fig. 1a, b) and hippocampus (Supplementary Fig. 1a, b), which confirmed the presence of extranuclear FUS clusters along dendrites and axons (identified with microtubule-associated protein 2 (MAP2) and phospho-neurofilament (PNF), respectively) and association with synaptic markers (Synapsin 1 and postsynaptic density protein 95 (PSD95)). To determine the precise subsynaptic localization of FUS, we used superresolution microscopy on mouse hippocampal and cortical synapses to explore the distribution of FUS between excitatory and inhibitory synapses (Fig. 1c). Stimulated emission depletion (STED) microscopy was used to precisely determine the localization of FUS clusters compared to synaptic markers; vesicular GABA transporter (VGAT) was used as a marker for inhibitory synapses[46] and PSD95 for excitatory synapses[47]. Image analysis was used to calculate the distance of the closest neighbor (Supplementary Fig. 1c). Only FUS clusters within 200 nm from a synaptic marker were considered for this analysis. Our results showed that synaptic FUS preferentially associates with excitatory synapses, of which 46% contained FUS, while only 20% of analyzed inhibitory synapses showed FUS positivity (t-test, $p = 0.0016$; Fig. 1d).

To better define the precise localization of FUS within the synapse, cortical, and hippocampal primary cultures were immunolabeled for FUS along with pre- and postsynaptic markers (Fig. 1e and Supplementary Fig. 1d, e) and their relative distance was analyzed. At the presynapse, Synapsin 1 was used to label the vesicle reserve pool[48], and Bassoon to label the presynaptic active zone[49]. At the postsynaptic site, GluN2B, subunit of N-methyl-D-aspartate (NMDA) receptors, and GluA1, subunit of α-amino-3-hydroxy-5-methyl-4-isoxazolepropionic acid (AMPA) receptors, were used to label glutamatergic synapses. PSD95 was used to label the postsynaptic density zone[47]. Distribution of FUS at the synapse showed a closer association with Synapsin 1 compared to Bassoon, GluA1, Binding immunoglobulin protein (BiP), an endoplasmic reticulum (ER) marker, and GluN2B (Supplementary Fig. 1f, g). FUS also appeared to be closer to Bassoon compared to PSD95 (Supplementary Fig. 1f, g). A subset of FUS was also localized at the spine (Fig. 1e). To refine the precise localization of FUS, the

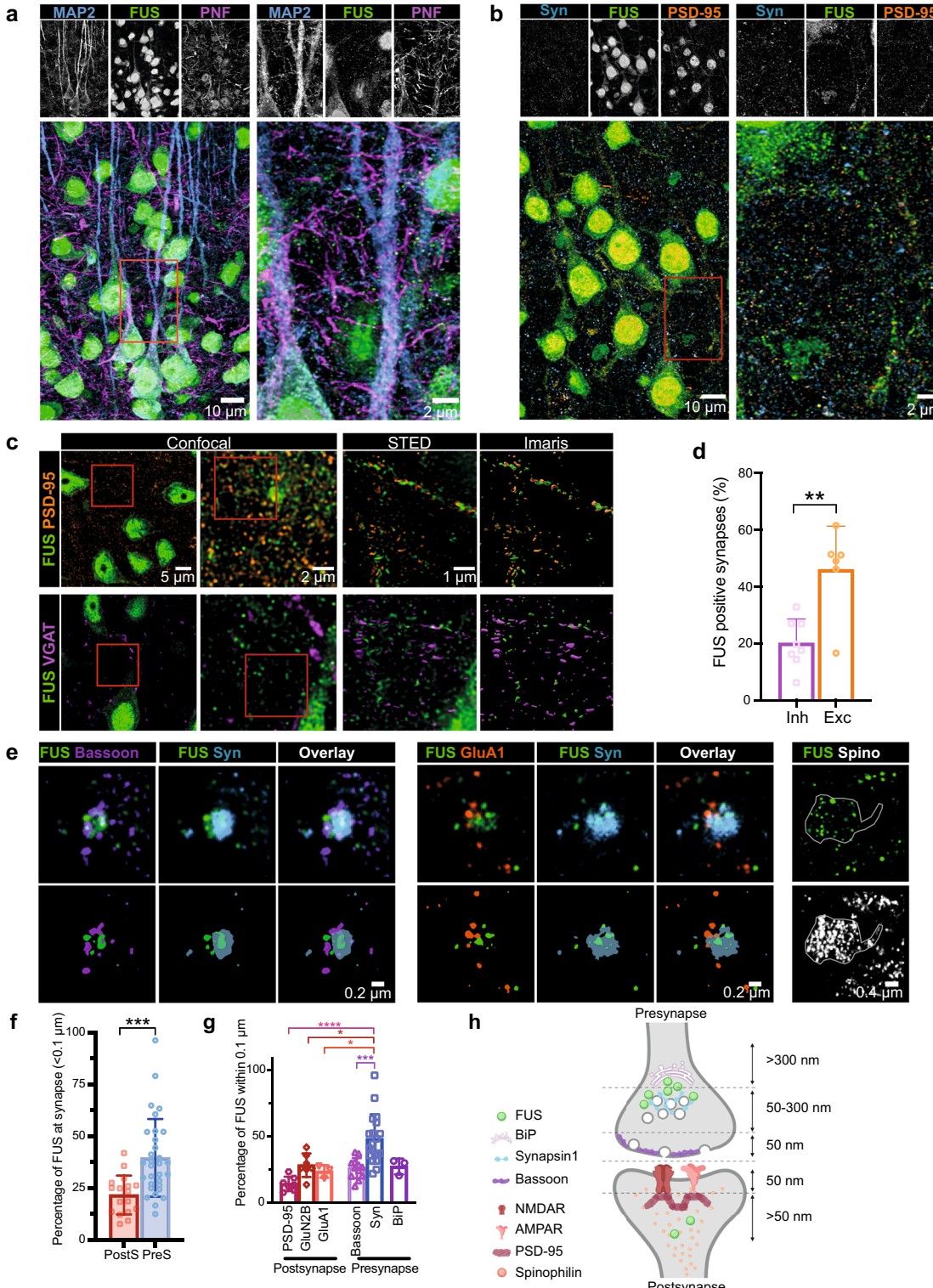

relative proportion of FUS within 100 nm was compared for each marker. Our results showed a preferential FUS localization at the presynaptic site (Fig. 1f; $t$-test, $p = 0.0006$), in accordance with previously reported data[25,27]. Within the presynaptic site (Fig. 1g), FUS was significantly enriched in the Synapsin-positive area (One-way ANOVA, $p < 0.0001$, posthoc Tukey, Syn1 vs. PSD95, $p < 0.0001$; Syn1 vs. GluN2B, $p = 0.0157$; Syn1 vs. GluA1, $p = 0.454$; Syn1 vs. Bassoon, $p = 0.0005$). However, no significant difference was found with the ER marker, suggesting that FUS could be localized between Synapsin 1 and ER at the presynapse

(Fig. 1h). These results are in line with the previously published localization of FUS within 150 nm from the active presynaptic zone[27], but highlight the presence of FUS also at the postsynaptic site, potentially explaining the apparently contradictory results of previous studies[20,28].

**Identification of synaptic RNA targets of FUS**. The role of FUS in the nucleus has been well studied and previously published CLIP-seq data identified FUS binding preferentially on pre-

**Fig. 1 FUS is enriched at the presynaptic compartment. a** Confocal images showing the distribution of Fused in sarcoma (FUS) (green) in the pyramidal layer of the retrosplenial cortical area along with microtubule-associated protein 2 (MAP2) (blue) and phospho-neurofilament (PNF) (magenta). Left panel shows the overview and the right panel the zoomed-in area labeled with the red box on the left panel. $n = 5$ independent mice. **b** Similar confocal images showing FUS (green) along with postsynaptic density protein (PSD95) (orange) and Synapsin 1 (Syn, blue). $n = 5$ independent mice. **c** Synaptic localization of FUS was assessed by STED microscopy using excitatory (PSD95) and inhibitory (vesicular GABA transporter (VGAT)) markers for synapses. 60 μm brain sections were analyzed and the distance between FUS and the synaptic markers was analyzed using Imaris. $n = 5$ independent mice. **d** Bar graph representing the percentage of synapses within 200 nm of FUS clusters and showing an enrichment of FUS at the excitatory synapses. $n = 5$ independent mice for both VGAT and PSD analysis. For VGAT (4 fields of view in the PTLp cortical area with $n = 286$ VGAT (+) synapses analyzed,), for PSD (2 fields of view in the PTLp cortical area with $n = 193$ PSD (+) synapses analyzed, Two-tailed, unpaired $t$-test, $p$ value = 0.0016. **e** dSTORM was used to explore the precise FUS localization within the synapse, using primary culture. Bassoon and Synapsin 1 (Syn) were used to label the presynaptic compartment and GluN1 (subunit of NMDAR), GluA1 (subunit of AMPAR) and PSD95 were used to label the postsynapse. Spinophilin (Spino) was used to label the spines. $n = 3$ independent neuronal cultures. **f** Bar graph representing the percentage of FUS localized within 100 nm from presynaptic or postsynaptic markers. Images were acquired in 2D and represent only one plane. Each punctum represents one imaged field of view. $n = 3$ independent neuronal culture, with $n = 17$ fields of view analyzed for postsynaptic markers and 31 fields of view analyzed for presynaptic markers. $p$ value = 0.0006, two-tailed, unpaired $t$-test. **g** Bar graph representing the distribution of FUS in the synapse. $n = 3$ independent cultures, $n = 7$ fields of view for PSD95 and GluN2B (subunit of NMDAR), $n = 3$ fields of view for GluA1, $n = 12$ fields of view for Bassoon, $n = 19$ fields of views for Synapsin 1 and $n = 3$ fields of view for endoplasmic reticulum (ER) marker (BiP). $p$ value = <0.0001, One-way ANOVA with Tukey's multiple comparisons post hoc test. **h** Schematic summarizing the FUS localization within the synapse. Graph bar showing mean + SD. *$p > 0.05$, **$p > 0.01$, ***$p > 0.001$, ****$p > 0.0001$.

mRNA, suggesting that these binding events occur in the nucleus[22,50–53]. Given the confirmed synaptic localization of FUS (Fig. 1), we wondered if a specific subset of synaptic RNAs is directly bound and regulated by FUS in these compartments. Since synapses contain few copies of different RNAs and only a small fraction of the total cellular FUS is synaptically localized, RNAs specifically bound by FUS at the synapses are likely missed in CLIP-seq datasets from total brain. Therefore, we biochemically isolated synaptoneurosomes that are enriched synaptic fractions from mouse cortex to identify synapse-specific RNA targets of FUS. Electron microscopy analysis confirmed the morphological integrity of our synaptoneurosome preparations, which contained intact pre- and postsynaptic structures (Fig. 2a). Immunoblot showed an enrichment of synaptic markers (PSD-95, phospho-calcium/calmodulin-dependent kinase IIα (p-CAMKIIα), GluN2B, GluA1, Synaptosomal-associated protein 25 (SNAP25), Neurexin 1 (NRXN1), absence of nuclear proteins (Lamin B1, Histone H3; Fig. 2b and Supplementary 2a) and presence of FUS in the synaptoneurosomes (Fig. 2b and Supplementary 2b). In addition, quantitative reverse transcription polymerase chain reaction (qRT-PCR) analysis showed enrichment of selected synaptic mRNAs (Fig. 2c).

Following a previously published method[22,54], we used ultraviolet (UV) crosslinking on isolated synaptoneurosomes and total cortex from 1-month-old wild-type C57Bl/6 mice to stabilize FUS-RNA interactions and to allow stringent immunoprecipitation of the complexes (Supplementary Fig. 2c). As FUS is enriched in the nucleus and only a small fraction of the protein is localized at the synapses, we prepared synaptoneurosomes from cortices of 200 mice to achieve sufficient RNA levels for CLIP-seq library preparation. The autoradiograph showed an RNA smear at the expected molecular weight of a single FUS molecule (70 kDa) and lower mobility complexes (above 115 kDa) that may correspond to RNAs bound by more than one FUS molecule or a heterogeneous protein complex (Fig. 2d). No complexes were immunoprecipitated in the absence of UV cross-linking or when using non-specific IgG-coated beads. The efficiency of immunoprecipitation was confirmed by depletion of FUS in post-IP samples (Supplementary Fig. 2d). Finally, RNAs purified from the FUS-RNA complexes of cortical synaptoneurosomes and total cortex at 70 kDa were sequenced and analyzed. We obtained 29,057,026 and 27,734,233 reads for the total cortex and cortical synaptoneurosome samples, respectively. In total, 91% of the total cortex and 66% of the synaptoneurosome reads could be mapped to a unique location in the mouse reference genome (GRCm38;

Supplementary Fig. 2e). After removing PCR duplicates, we identified peaks using CLIPper[55], resulting in 619,728 total cortex and 408,918 synaptoneurosome peaks.

We designed a pipeline to identify FUS binding sites that are specific for the synapse and not bound in the nucleus. Before comparing the peaks in the two samples, we normalized the data to correct for different sequencing depths and signal-to-noise ratios[56] (see Methods section). This is especially important, because the synaptoneurosome sample should contain only a subset of the FUS targets from total cortex. We wanted to filter the predicted peaks of the synaptoneurosome sample to identify genomic regions with high log2 fold-change between synaptoneurosome and total cortex. Peaks with low number of reads (or no reads) in the total cortex, but high read coverage in the synaptoneurosomes correspond to regions that are putatively bound by FUS in the synapse. However, the observable number of reads per RNA in each sample strongly depends on gene expression and the number of localized RNA copies. Therefore, we did not want to use a simple read count threshold to filter and identify synapse specific peaks. Instead, we fit a count model and computed peak-specific $p$-values to test for differences between the synaptoneurosome and total cortex CLIP-seq enrichment (Fig. 2e).

We ranked the peaks by $p$-values and used a stringent cutoff of 1e-5 (Fig. 2f) to ensure enrichment of synaptic FUS targets. Indeed, the resulting peaks were largely devoid of intronic regions, but were enriched in exons and 3′UTRs, as was expected for synaptic FUS targets, which are mature and fully processed RNAs (Fig. 2f and Supplementary Fig. 2h). The same normalization and filtering of CLIPper peaks identified in the total cortex highlighted RNAs primarily bound by FUS in the nucleus, where the vast majority of FUS protein resides (Supplementary Fig. 2f). After selecting an equal number of top peaks as obtained for the synaptoneurosome sample (1560 peaks in 517 genes), corresponding to a $p$-value cutoff of 0.0029 (Supplementary Fig. 2g), we confirmed the previously reported[22] preferential binding of FUS within intronic regions of pre-mRNAs (Fig. 2g and Supplementary Fig. 2i).

Despite the strong reduction of intronic binding in our synaptoneurosome CLIP-seq, 227 peaks (corresponding to 13% of all synaptic peaks) were found within regions annotated as introns. We first considered that these were weak peaks, potentially inaccurately called by our pipeline. However, when we compared the maximal peak height of intronic peaks to those found in coding regions, we found no significant difference

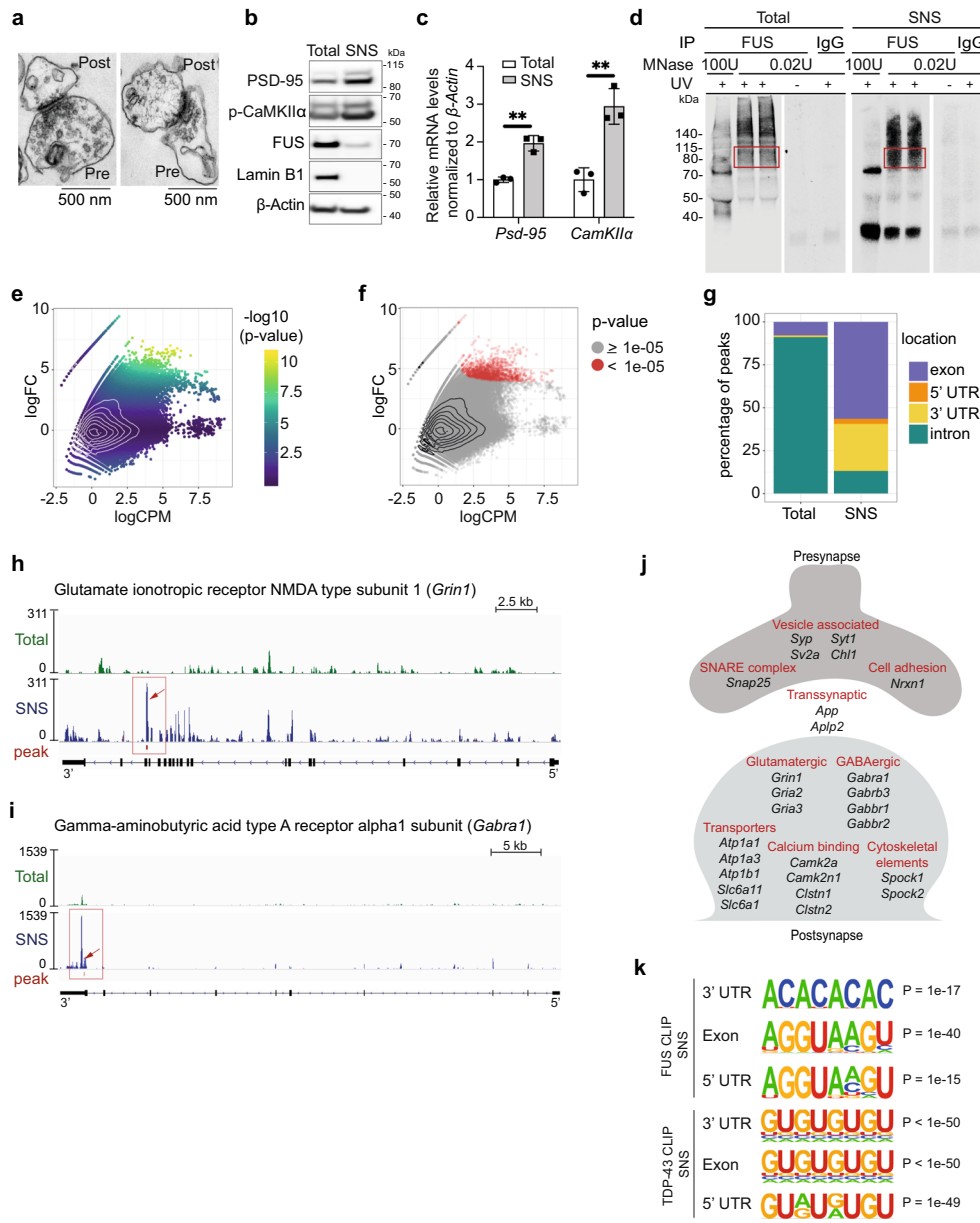

**Fig. 2 CLIP-seq on cortical synaptoneurosomes identified FUS-associated pre- and postsynaptic RNAs. a** Electron microscopic images of synaptoneurosomes (SNS) from mouse cortex showing intact pre- and postsynaptic compartments. $n = 2$, independent experiments. **b** Western blot of synaptic proteins (postsynaptic density protein (PSD95), phospho-calcium/calmodulin-dependent kinase IIα (p-CamKII)), nuclear protein (Lamin B1), and Fused in sarcoma (FUS) in total cortex and synaptoneurosomes. $n = 6$ biological replicates. Source data are provided as a Source Data file. **c** qPCR shows enrichment of *PSD95* (p-value = 0.0015), and *CamKII* mRNAs (p-value = 0.0040) in synaptoneurosomes compared to the total cortex. Bar graphs represent mean ± SD. Each dot represents total and SNS samples prepared from three independent mice. $n = 3$, **$p < 0.01$, two-tailed, unpaired $t$-test. **d** Autoradiograph of FUS-RNA complexes immunoprecipitated from total cortex and synaptoneurosomes and trimmed by different concentrations of micrococcal nuclease (MNase). The red box indicates the excised part used for preparing CLIP libraries. Cortices from 200 mice were used to prepare synaptoneurosomes and two mice for the total cortex sample. Due to the necessity of this significant upscaling, library preparation and sequencing was not repeated ($n = 1$). However, immunoprecipitation and autoradiograph has been repeated several times to confirm reproducibility. **e** MA-plot of CLIPper peaks predicted in the synaptoneurosome CLIP-seq sample. logCPM is the average log2CPM of each peak in the total cortex and synaptoneurosome sample and logFC is the log2 fold-change between the number of reads in the SNS and total cortex sample. The row of points in the upper left corner are peaks with zero reads in the total cortex sample. P-values shown in **e**, **f** are computed by likelihood-ration test and unadjusted for multiple comparisons. **f** Same MA-plot as in **e** showing the selected, synaptoneurosome-specific peaks (p-value cutoff of 1e-05) in red. **g** Bar plot with the percentage of synaptoneurosome and total cortex-specific peaks located in exons, 5'UTRs, 3'UTRs or introns. FUS binding in *Grin1* (**h**), *Gabra1* (**i**) in total cortex (green), and SNS (blue). **j** Schematic with the cellular localization and function of some of the selected FUS targets. **k** Predicted sequence motifs (HOMER) in windows of size 41 centered on the position with maximum coverage in each peak. Each set of target sequences has a corresponding background set with 200,000 sequences without any CLIP-seq read coverage (they are not bound by FUS/TDP-43). We note that these are all FUS motifs that were marked as true positives by HOMER and that occur in more than 3% of the target sequences, while the top motif is shown for TDP-43. P-values are reported by HOMER (one-sided hypergeometric test).

(Supplementary Fig. 3a), confirming the validity of our pipeline. We then utilized Ribo-Zero RNA sequencing data from total mouse cortex or synaptoneurosome preparations to explore the detectable levels of nuclear RNA in each sample. The nuclear-enriched non-coding RNA *Malat1* (Metastasis Associated Lung Adenocarcinoma Transcript 1)[57–59] was strongly detected in total cortex samples, with only trace coverage seen in synaptoneurosomes (Supplementary Fig. 3b). Similarly, while in the total cortex samples pre-mRNA detection was evident due to intronic coverage, these were clearly missing in synaptoneurosomes (Supplementary Fig. 3c), suggesting that nuclear contamination, if any, was minimal, in line with our previous analysis (Fig. 2a–d). Nevertheless, we cannot exclude the possibility that some of these intronic binding sites might be carried over from the nucleus. This might be true, for example, in the case of *Slc9a9* (Solute carrier family 9 member 9), which is strongly bound by nuclear FUS on multiple intronic locations (Supplementary Fig. 3d), even though the single synaptoneurosome-specific peak in *Slc9a9* RNA may also reflect binding to a yet unannotated non-coding RNA. This is likely the case for some other peaks annotated as intronic in synaptoneurosomes. An example is the binding identified within the last intron of *Peak1*, which overlaps with the microRNA *Gm24270*. The latter is consistently detected in all samples, including total cortex and synaptoneuromes in both Ribo-Zero and poly(A) RNA-seq libraries. The position of synaptic FUS binding indicates association with the pre-miRNA of *Gm24270* (Supplementary Fig. 3e).

The final list of synapse-specific FUS binding sites consists of 1560 peaks in 307 RNAs (Supplementary data 1), primarily localized to exons and 3′UTRs of RNAs specific to the synapses. Among those, FUS peaks on exon 18 of *Grin1* (Glutamate ionotropic NMDA type subunit 1) and the 3′UTR of a long isoform of *Gabra1* (Gamma aminobutyric acid receptor subunit alpha-1) were exclusively detected in synaptoneurosomes, but not in total cortex (Fig. 2h, i). Direct binding of FUS to 3′UTR and exonic regions of its targets suggests a potential role in regulating RNA transport, local translation and/or stabilization.

**Synaptic FUS RNA targets encode essential protein components of synapse.** We then wondered if the 307 synaptic FUS target RNAs were collectively highlighting any known cellular localization and function. Most RNAs are localized to either the pre- or postsynapse or they are known astrocytic markers (Fig. 2j). Among those are RNAs encoding essential protein members of glutamatergic (*Grin1*, *Gria2*, *Gria3*) and GABAergic synapses (*Gabra1*, *Gabrb3*, *Gabbr1*, *Gabbr2*), transporters, as well as components of the calcium signaling pathway, which are important for plasticity of glutamatergic synapses. An over-representation analysis (ORA) comparing the synaptic FUS targets to all synaptic RNAs detected in cortical mouse synaptoneurosomes by RNA-seq (logCPM >1, 1-month-old mice), revealed that FUS targets were enriched for synaptic – both pre- and postsynaptic – localization. Synaptic FUS target RNAs were enriched for gene ontology (GO) categories, such as transport, localization, and trans-synaptic signaling, as well as signaling receptor binding and transmembrane transporter activity (Supplementary Fig. 2j). Our data suggests that FUS may play an important role in maintaining synaptic integrity and organization.

**FUS binds GU-rich and AC-repeat sequences at the synapse.** While FUS has been shown to be a relatively promiscuous RNA-binding protein, preference towards GU-rich motifs has been reported in previous CLIP-seq studies[22,38,40,41], a binding mediated via its ZnF domain[42]. To understand if FUS binding to synaptic RNA targets follows the same modalities as its nuclear targets, we explored the sequence specificity of FUS in the synapse and predicted motifs with HOMER[60] in the FUS peak sequences of cortical synaptoneurosomes. The sequences of the synaptic FUS peaks in exons and 5′ UTRs revealed a "AGGUAAGU" motif known to be bound by TAF15[61] which was only found in 11% and 6% of the peaks, respectively. Moreover, we identified a novel AC-repeat binding motif in 7% of synaptic 3′ UTR FUS targets (Fig. 2k), compatible with the previously reported binding of the RNA recognition motif (RRM) of FUS within stem loop regions that contain AC motifs[42,43].

To validate our findings on the synaptic RNA FUS binding, we performed CLIP-seq for TDP-43, following the identical protocol for preparing the libraries from synaptoneurosomes that we used for FUS. We reasoned that TDP-43 is an ideal control for our method because, like FUS, its synaptic levels are considerably lower than its nuclear levels, but, unlike FUS, TDP-43 has a well characterized sequence specificity to GU repeats[54,62]. Our analysis identified the previously published GU-repeat as a significantly enriched TDP-43-binding motif (Fig. 2k). This result indicates that our SNS CLIP-seq protocol yields specific RBP-RNA-binding profiles and that the motifs identified in SNS FUS CLIP-seq are valid.

**Increased synaptic localization of mutant FUS protein in $Fus^{\Delta NLS/+}$ mice.** To explore synaptic impairment associated with FUS mislocalization, we used the $Fus^{\Delta NLS/+}$ mouse model[44]. This mouse model shows partial cytoplasmic mislocalization of FUS due to a lack of the nuclear localization (NLS) in one of the *Fus* alleles, closely mimicking ALS-causing mutations reported in patients. Taking advantage of two antibodies that recognize either total FUS (both full length and mutant) or only the full-length protein (Fig. 3a), we assessed FUS protein levels in synaptoneurosomes isolated from $Fus^{\Delta NLS/+}$ mice and wild type ($Fus^{+/+}$) of 1 and 6 months of age. We detected a ~3-fold increase in total FUS levels in synaptoneurosomes from $Fus^{\Delta NLS/+}$ at both ages compared to $Fus^{+/+}$ (Fig. 3b, c and Supplementary Fig. 4a, b). However, full-length FUS levels were decreased in synaptoneurosomes of $Fus^{\Delta NLS/+}$ compared to $Fus^{+/+}$ indicating that the truncated FUS protein is misaccumulated at the synaptic sites of $Fus^{\Delta NLS/+}$ mice. Confirming our biochemical evidence, immunofluorescence analyses of $Fus^{\Delta NLS/+}$ mice showed higher levels of FUS in dendritic compartments of CA1 pyramidal cells. $Fus^{+/+}$ mice at both 1 month (Supplementary Fig. 4c, d) and 6 months of age (Fig. 3d, e) showed prominent expression of FUS in the nucleus. High magnification images highlighted the presence of FUS at the synapses, identified by co-labeling with Synapsin1. $Fus^{\Delta NLS/+}$ mice at 1 (Supplementary Fig. 4c, d) and 6 months of age (Fig. 3d, e) showed higher levels of FUS within the dendritic tree (identified with MAP2) and at the synapse compared to $Fus^{+/+}$ mice, confirming our previous quantifications by immunoblot.

To identify if FUS accumulation alters the levels of specific synaptic proteins, we performed proteomic analysis of synaptoneurosomes from 6-month-old $Fus^{\Delta NLS/+}$ and $Fus^{+/+}$ mice (Supplementary Fig. 4e). This analysis confirmed a significant ~3-fold increase of FUS in the synaptic sites of $Fus^{\Delta NLS/+}$ mice, in full agreement with our biochemical and image analysis (Fig. 3a–d). However, while some trends were detected, no other significant changes were found at this time point, a result that was confirmed in three independent mass spectrometry experiments (not shown). A possible reason for the lack of protein level alterations at 6 months may be that changes are too mild to be detected by a mass spectrometry approach. However, we cannot exclude that proteome changes occur at the synaptic sites at later

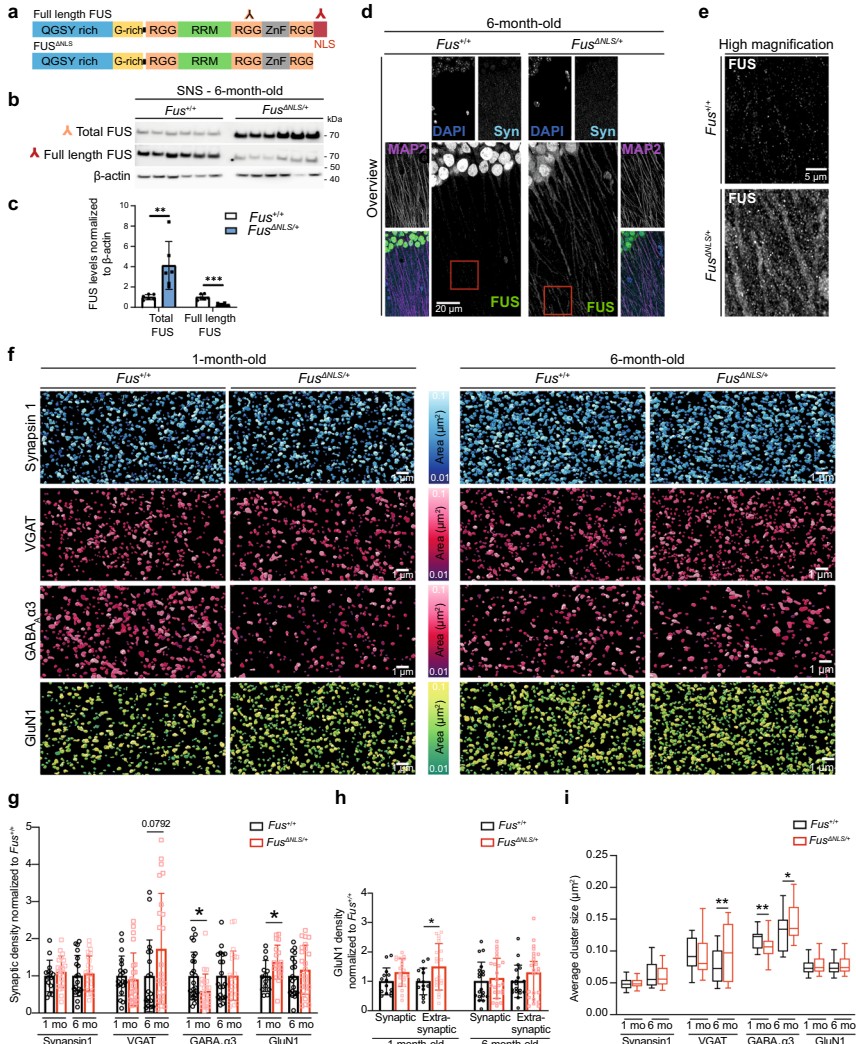

**Fig. 3 Increased synaptic FUS localization in *Fus*[ΔNLS/+] mice is accompanied by alterations in GABAergic synapses. a** Schematic showing specificity of antibodies used for western blot against protein domains of Fused in sarcoma (FUS). **b** Western blot of total FUS, full-length FUS and actin in synaptoneurosomes isolated from *Fus*[+/+] and *Fus*[ΔNLS/+] mice at 6 months of age. Source data are provided as a Source Data file. **c** Quantification of total FUS (*p*-value = 0.0091) and full-length FUS levels (*p*-value = 0.001) in synaptoneurosomes from *Fus*[+/+] and *Fus*[ΔNLS/+] at 6 months of age. Error bars represent mean ± SD. Each dot represents samples prepared from six different mice. *n* = 6, **$p < 0.01$, ***$p < 0.001$, two-tailed, unpaired *t*-test. **d** Confocal images of the hippocampal CA1 area from 6-month-old mice showing higher level of FUS in the dendritic tree and synaptic compartment in *Fus*[ΔNLS/+] mouse model. On the top, low magnification pictures show the dendritic area of pyramidal cells stained with FUS (green), microtubule-associated protein 2 (MAP2) (dendritic marker, magenta), Synapsin 1 (Syn, synaptic marker, Cyan), and DAPI (Blue). Red box indicates the area imaged in the high magnification images below. *n* = 5 independent *Fus*[+/+] and *Fus*[ΔNLS/+] mice. **e** Higher magnification equivalent to the area highlighted in red in **d**. **f** Representative images of staining using synaptic markers Synapsin 1, Vesicular GABA transporter (VGAT), postsynaptic GABAergic receptors containing α3 subunit (GABA_Aα3), and GluN1 (NMDAR submit) in *Fus*[+/+] and *Fus*[ΔNLS/+] at 1 and 6 months of age. Images were generated with Imaris and display volume view used for quantification with statistically coded surface area. Density and cluster area were analyzed. *n* = 6 independent *Fus*[+/+] and *Fus*[ΔNLS/+] mice at 1 month of age. At 6-month-old, *n* = 5 independent mice for *Fus*[+/+], *n* = 6 for *Fus*[ΔNLS/+]. **g** Graph bar representation of the synaptic density of Synapsin 1, VGAT, GABA_Aα3, and GluN1 from *Fus*[+/+] and *Fus*[ΔNLS/+] at 1 and 6 months of age. Graph bar showing mean ± SD. *$p < 0.05$, two-tailed unpaired *t*-test. Graphs are extracted from the same analysis shown in Supplementary Fig. 3e, f. The statistical analysis can be found in Supplementary data 2. For 1-month-old group, *n* = 6 independent mice for *Fus*[+/+] and *Fus*[ΔNLS/+]. For 6-month-old group, *n* = 5 independent mice for *Fus*[+/+], *n* = 6 for *Fus*[ΔNLS/+]. For Synapsin 1 at 1-month-old, *n* = 14 fields of view for *Fus*[+/+] and 866013 synapses analyzed, 20 fields of view for *Fus*[ΔNLS/+] with 1,475,455 synapses analyzed. For Synapsin 1 at 6-month-old *n* = 19 fields of view for *Fus*[+/+] and 1590276 synapses analyzed, 24 fields of view for *Fus*[ΔNLS/+] with 2,138,361 synapses analyzed. For VGAT at 1-month-old, *n* = 19 fields of view for *Fus*[+/+] with 713,918 synapses analyzed, 23 fields of view for *Fus*[ΔNLS/+] with 782790 synapses analyzed. For VGAT at 6-month-old, *n* = 18 fields of view for *Fus*[+/+] with 553020 synapses analyzed, 24 fields of view for *Fus*[ΔNLS/+] with 1,273,325 synapses analyzed. For GABA_Aα3 at 1-month-old, *n* = 24 fields of view for *Fus*[+/+] with 1,062,095 synapses analyzed, 24 fields of view for *Fus*[ΔNLS/+] with 634,958 synapses analyzed. For GABA_Aα3 at 6-month-old, *n* = 20 fields of view for *Fus*[+/+] with 726,881 synapses analyzed, 22 fields of view for *Fus*[ΔNLS/+] with 804,732 synapses analyzed. For GluN1 at 1-month- old, *n* = 14 fields of view for *Fus*[+/+] with 704,322 synapses analyzed, 20 fields of view for *Fus*[ΔNLS/+] with 1,379,868 synapses analyzed. For GluN1 at 6-month-old, *n* = 19 fields of view for *Fus*[+/+] with 1,267,271 synapses analyzed, 24 fields of view for *Fus*[ΔNLS/+] with 1,866,397 synapses analyzed. **h** Colocalization analysis of GluN1 with Synapsin 1 to identify synaptic NMDAR (GluN1) and extrasynaptic NMDAR. Results were normalized by the control of each group. Graph bar showing mean ± SD. *$p < 0.05$., two-tailed unpaired *t*-test. '*n*' values are as indicated in **g**. **i** Box and Whiskers representation (centered on the median with min to max value) of the average cluster area for each marker (Synapsin 1, VGAT, GABA_Aα3, and GluN1) from 1-month and 6-month-old *Fus*[+/+] and *Fus*[ΔNLS/+] mice. Box showing Min to Max, *$p < 0.05$ **$p < 0.01$, two-tailed, unpaired *t*-test. '*n*' values are as indicated in **g**. Graphs are extracted from the same analysis shown in Supplementary Fig. 3f–i. The statistical analysis can be found in Supplementary data 3.

time points and closer to the age of disease onset. Another important consideration is that our preparation of synaptoneurosomes contains total cortical tissue, consisting of different neuronal types. Therefore, subtle changes in protein levels occurring in only one type of synapse (inhibitory or excitatory) might not be detectable. To explore this possibility, we used image analysis to evaluate the synaptic integrity, an approach that allows for focusing on specific synaptic subtypes.

**Dysregulation of inhibitory synapses in $Fus^{\Delta NLS/+}$ mouse model.** To explore a possible synaptic disorganization associated with mislocalization and accumulation of FUS at synaptic sites, we performed synaptic density and size analyses. Based on evidence that the hippocampal/prefrontal cortex connectome participates in memory encoding and recalling[63] and that CA1 hippocampal excitatory and inhibitory synapses are highly similar to the cortical synapses[64–67], we explored the possible synaptic changes triggered by FUS mislocalization in the CA1 hippocampal region. We analyzed both $Fus^{+/+}$ and $Fus^{\Delta NLS/+}$ mice, using presynaptic and postsynaptic markers. Density and area analyses were performed as shown in Supplementary Fig. 4f. At the presynapse, we quantified the density of the SNARE associated protein SNAP25[68] (synaptic RNA target of FUS) and the presynaptic active zone marker Bassoon[49]. The density of inhibitory synapses was assessed using VGAT[46] (presynaptic). At the postsynapse, we quantified the density of postsynaptic glutamatergic receptor GluN1[69] (synaptic RNA target of FUS and obligatory subunit of all NMDAR) and GluA1[70] (obligatory subunit of AMPAR), as well as postsynaptic GABAergic receptors containing α1 subunit (GABA$_A$α1; synaptic RNA target of FUS) and α3 (GABA$_A$α3)[71]. We also assessed the number of active excitatory synapses using phospho-CaMKII (p-CaMKII) as well as functional inhibitory synapses using Gephyrin[72].

At 1 month of age in $Fus^{\Delta NLS/+}$ mice, we did not observe significant changes at the presynaptic site, suggesting a normal axonal and axon terminal development and functions. However, at the postsynaptic sites, we observed a significant increase of NMDAR ($p = 0.0219$) and a significant decrease of GABA$_A$α3 receptors ($p = 0.0156$; Fig. 3f, g, Supplementary Fig. 4g, and Supplementary data 2). Moreover at 1 month of age, $Fus^{\Delta NLS/+}$ mice showed significantly more NMDAR located at the extrasynaptic site ($p = 0.0433$; Fig. 3h). Interestingly, the size of the GABA$_A$α3 clusters was significantly decreased in $Fus^{\Delta NLS/+}$ mice ($p = 0.0053$) at 1 month of age (Fig. 3f, i, Supplementary Fig. 4i, and Supplementary data 3). We did not record changes in the number of Synapsin1, Bassoon, SNAP25, VGAT, GluA1, GABA$_A$α1, Gephyrin, or pCaMKII, suggesting either an increase of silent synapses, immature synapses or an increase of the number of NMDAR in the dendritic shaft together with a decrease of GABA$_A$α3 synaptic clustering. These results suggested a hyperexcitability profile during developmental stages.

At 6 months of age, we did not observe significant changes in the density of pre- or postsynaptic markers (Fig. 3f, g and Supplementary Fig. 4h), suggesting a normal maturation of the synaptic network despite developmental synaptic dysregulation described above. However, SNAP25 ($p = 0.085$) and VGAT ($p = 0.0792$) trended towards an increased density, suggesting a potential alteration at inhibitory presynaptic sites (Supplementary Fig. 4h and Supplementary data 2). This interpretation was confirmed by an increase of the area of the presynaptic marker VGAT ($p = 0.0028$) and of the size of GABA$_A$α3 clusters at the postsynaptic site ($p = 0.0166$; Fig. 3i, Supplementary Fig. 4j, and Supplementary data 3), while GluN1 clusters appeared unaffected. Increase in VGAT suggested an elevated number of presynaptic GABAergic vesicles, which was confirmed by EM

analyses in older mice[45]. Correlatively, increase of GABA$_A$α3 cluster size suggested an increase in the trafficking of GABA$_A$R at the postsynaptic site. This occurred, however, without an increase of the anchoring protein Gephyrin, suggesting unstable structure of the inhibitory postsynaptic sites. Altogether, our results show alterations of both glutamatergic and GABAergic synapses during developmental synaptogenesis (1 month), while only GABAergic synapses appeared affected at a later time point (6 months). This suggests a potential role for FUS in synaptogenesis and network wiring and synaptic maintenance, with a selective exacerbation of inhibitory synaptic defects with age. These early synaptic changes mechanistically explain the behavioral dysfunctions that these mice develop[45]. In particular, $Fus^{\Delta NLS/+}$ mice displayed increased locomotor activity at 4 months of age, while motor symptoms start at 10 months of age[44]. Moreover, they exhibit impairments in long-term memory consolidation and social inhibition starting from 10 months of age[45].

**$Fus^{\Delta NLS/+}$ mice show age-dependent synaptic RNA alterations.** FUS plays an essential role in RNA stabilization[23,24] and transport[20]. Therefore, we used RNA-seq to investigate the consequences of increased synaptic levels of mutant FUS in $Fus^{\Delta NLS/+}$ mice (Fig. 4a). We isolated RNA from six biological replicates of synaptoneurosomes and paired total cortex samples from $Fus^{+/+}$ and $Fus^{\Delta NLS/+}$ mice at 1 and 6 months of age and prepared poly(A)-selected libraries for high-throughput sequencing. As a control, we also sequenced the nuclear fraction from four biological replicates of $Fus^{+/+}$ mice at 1 month of age. For quality control, we computed principal components of all samples and all expressed genes (methods) and found a clustering by sample type and age (Supplementary Fig. 5b, c).

We compared the expressed genes in our synaptoneurosomes (15,087 genes) with the forebrain synaptic transcriptome[73] (14,073 genes) and found the vast majority of detected RNAs (13,475) to be identical between the two studies (Supplementary Fig. 5a). The small differences in the two transcriptomes can be explained by differences in the synaptoneurosome protocols and the brain region (frontal cortex versus forebrain).

We conducted four differential gene expression analyses, comparing $Fus^{\Delta NLS/+}$ to $Fus^{+/+}$ separately for the total cortex and synaptoneurosomes at both time points (for full lists see Supplementary data 4–7). A false discovery rate (FDR) cutoff of 0.05 was used to define significant differential expression. Only three and five mRNAs were differentially expressed (DE) in the $Fus^{\Delta NLS/+}$ samples of the total cortex at 1 and 6 months of age, respectively (Supplementary Fig. 5f and Supplementary data 4 and 5). However, in the synaptoneurosomes, we identified 11 and 594 mRNAs differentially abundant at 1 and 6 months, respectively (Supplementary data 6 and 7). In all, 136 mRNAs were decreased and 485 mRNAs were increased in the synaptoneurosomes of $Fus^{\Delta NLS/+}$ mice at 6 months of age compared to $Fus^{+/+}$ mice (Fig. 4b). We performed an ORA on the sets of increased and decreased mRNAs and used the genes that are expressed in all $Fus^{+/+}$ SNS samples at 6 months as background. The significantly increased RNAs in $Fus^{\Delta NLS/+}$ mice at 6 months were enriched in GO categories such as synaptic signaling, intrinsic component of membrane and transporter activity (Supplementary Fig. 5d), while those that were decreased in abundance were associated with cytoskeletal organization and RNA metabolism (Supplementary Fig. 5e).

At 6 months of age, the log2 fold changes of the altered mRNAs are consistently negative or positive in all $Fus^{\Delta NLS/+}$ synaptoneurosome replicates (Fig. 4c). At 1 month of age, the log2 fold changes of the $Fus^{\Delta NLS/+}$ synaptoneurosome replicates are mostly neutral (white color on the heatmap) indicating that

alterations in RNA abundance are age-dependent and not detectable as early as 1 month of age. In the total cortical samples at 6 months of age, some of the replicates show a similar trend as the synaptoneurosome samples, but it seems that the effects cannot be detected because synaptic RNAs are too diluted (Fig. 4c). Overall, we found synapse-specific differential RNA abundance at 6 months in the $Fus^{\Delta NLS/+}$ mice, but not in total cortex.

The 136 decreased mRNAs at the synaptic sites included only 5 synaptic FUS targets, suggesting that the majority of these were not directly affected by the increased synaptic FUS localization. In contrast, there are 28 FUS targets among the 485 increased synaptic mRNAs, representing a significant enrichment ($p = 3.1e-09$ one-sided Fisher's exact test, Fig. 4d). While this significant enrichment suggests that FUS binding may mediate the synaptic accumulation of these mRNAs, further studies are required to fully characterize the functional implication of the direct FUS binding on its targets. Most of these 28 FUS-interacting mRNAs with increased accumulation show exonic FUS binding on our CLIP-seq analysis (Supplementary Figs. 6 and 7 and Supplementary data 1), with the exception of *Gria 3*, *Spock1*, and *Spock2* (Supplementary Fig. 7b, f, g), which are bound by FUS at their 3′ UTR. Altered FUS targets include RNAs encoding presynaptic vesicle associated proteins, trans-synaptic proteins, membrane proteins and receptors associated with glutamatergic and GABAergic pathways.

**Cytoplasmic FUS accumulation in $Fus^{\Delta NLS/+}$ mice increases the stability of synaptic mRNAs.** Since FUS primarily binds to exons and 3′UTR of its synaptic targets and the cytoplasmic accumulation of FUS alters synaptic RNA levels, we reasoned that FUS binding may directly affect the stability of some of its targets. While it has been shown before that knock down of FUS in primary cortical neurons[23] and human neuronal precursor cells[61] leads to altered stability of mRNAs, the effect of FUS accumulation on mRNA stabilization has never been addressed before. To unbiasedly investigate the effect of cytoplasmic FUS accumulation on mRNA stability, we performed poly(A) RNA-seq upon treating $Fus^{\Delta NLS/+}$ and $Fus^{+/+}$ primary neurons with a transcriptional inhibitor, actinomycin D. We sequenced four different time points: 0 h (no treatment), 8 h, 12 h, and 24 h post treatment (Fig. 5a). In a PCA plot, the samples clustered by time point with the controls (0 h) clearly separated from treated samples (Supplementary Fig. 5a).

We performed differential gene expression analysis comparing $Fus^{\Delta NLS/+}$ to $Fus^{+/+}$ samples at each time point to determine changes in mRNA half-life or decay over time (for full lists see Supplementary data 8). As expected, we found no significant changes between the two genotypes in the untreated samples (0 h; Fig. 5b). However, 8 h after actinomycin D treatment, 337 and 485 mRNAs decreased or increased in levels in $Fus^{\Delta NLS/+}$, respectively. Alterations in mRNA levels peaked at 12 h, with 2091 decreased and 2046 increased mRNAs in $Fus^{\Delta NLS/+}$ neurons, while by 24 h, mRNA levels between the two genotypes rectified with only 167 and 299 mRNAs found decreased or increased in $Fus^{\Delta NLS/+}$, respectively (Fig. 5b). The RNAs altered in $Fus^{\Delta NLS/+}$ at 8 h, 12 h, and 24 h, included synaptic mRNAs with FUS binding: 5 RNA targets decreased and 4 increased at 8 h, 28 RNA targets decreased and 48 increased at 12 h, and 3 RNA targets decreased and 3 increased at 24 h. We performed ORA to see if the increased and decreased genes are enriched for specific cellular localization or functionality using the expressed genes at 0 h as background. Importantly, mRNAs with increased levels in $Fus^{\Delta NLS/+}$ at 12 h were enriched for GO terms indicating synaptic

localization and function, including membrane, trans-synaptic signaling and neuron development (Fig. 5c), even though the stability assay was performed with total extracts and not synaptoneurosomes. In contrast, decreased mRNAs at 12 h were enriched for ribosomal localization and biogenesis, gene expression and translation, as well as RNA binding (Supplementary Fig. 9b). The enrichment of synaptic RNAs among the total more stable mRNAs suggested that FUS misaccumulation increases the stability of synaptic mRNAs.

We wondered how mRNA stability changed over time and if there were different classes of mRNAs that reacted differently following transcription inhibition. To answer these questions, we performed hierarchical clustering on the log2 fold changes of the two sets of increased and decreased mRNAs using a transformed Pearson correlation coefficient as distance measure and average linkage. We obtained three clusters with distinct patterns in the log fold changes for each set (Fig. 5d). The first cluster of the increased mRNAs showed increased stability at 12 h in $Fus^{\Delta NLS/+}$, the second cluster contained mRNAs with increased stability at 24 h, and the third cluster showed slightly increased stability at 8 h. The first cluster of the decreased mRNAs showed decreased stability at 12 h, cluster two contained mRNAs with decreased stability at 24 h, and the third cluster mRNAs with lower starting levels in $Fus^{\Delta NLS/+}$ and/or slightly decreased stability at 8 h followed by a strong increase at 12 h (Supplementary Fig. 9c). Most FUS targets among the increased mRNAs belong to cluster 1, a few to cluster 2, and there are no targets in cluster 3 (targets are highlighted in blue in Fig. 5d), indicating that a subset of RNAs bound by FUS are more stable at 12 and 24 h after blocking transcription.

**Exonic FUS binding on synaptic mRNAs correlates with increased stability.** When we compared the FUS binding distribution within the mRNAs altered at each time point, we observed that the RNA targets that showed increased stability in $Fus^{\Delta NLS/+}$ had FUS peaks mostly within exonic regions, while those with decreased stability had a higher proportion of 3′UTR peaks (Fig. 5e). The same exonic enrichment was also observed for mRNAs accumulated at 6-month-old $Fus^{\Delta NLS/+}$ mice. We tested these sets of FUS targets for association with the binding distribution and found a strong positive association of exonic binding with increased mRNAs (at 8 and 12 h post treatment) and with accumulated mRNAs at synaptoneurosomes of 6-month-old $Fus^{\Delta NLS/+}$ mice (Pearson's $\chi^2$ test: $p = 5.2e-05$ (8 h), $p = 0.003$ (12 h), $p = 1.9e-06$ (SNS accumulation)). The same trend can be observed when classifying the FUS targets according to where the majority of the peaks are located: targets with exonic binding have the highest proportion of accumulated mRNAs and less abundant mRNAs are bound at the 3′UTR (Fig. 5f). These observations are compatible with the notion that exonic, but not 3′UTR, FUS binding increases mRNA stability.

To determine if transcript length influences mRNA stability, we compared the length distribution of the longest transcript, coding region (CDS), 3′UTR and 5′UTR of the genes in decreased and increased sets. We found that decreased mRNAs were overall shorter than increased (Supplementary Fig. 9d). However, when we compared the length distributions between clusters within a gene set, we observed a negative correlation between transcript length and stability: shorter transcripts are more stable at 12 h and 24 h (cluster 1 and 2 of increased RNAs and cluster 3 of decreased mRNAs). This weak negative correlation between transcript length and stability has been observed for different organisms including yeast[74], *E. coli*, and humans[75,76], and it varies between cell types. Taken together, these results indicate

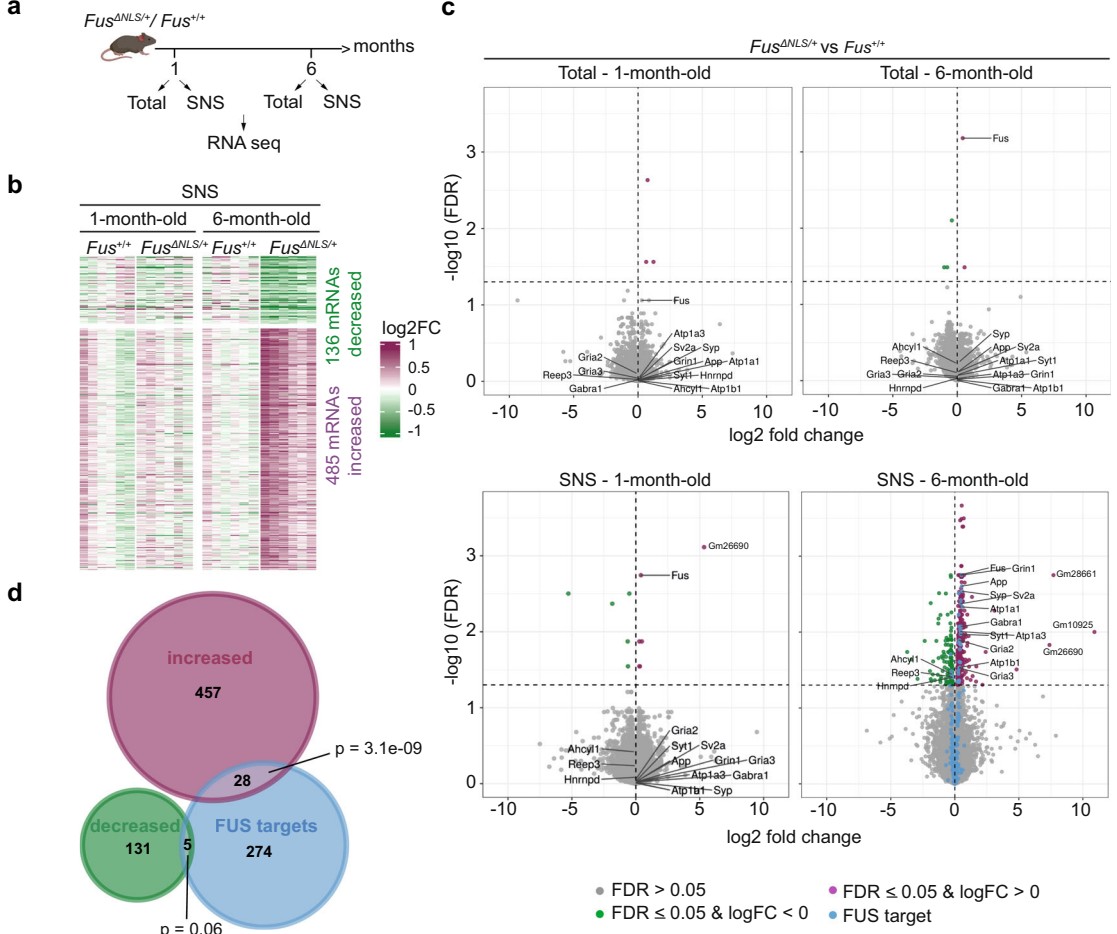

**Fig. 4 Age-dependent alterations in the synaptic mRNA profile of *Fus^{ΔNLS/+}* mouse cortex. a** Outline of the RNA-seq experiment. **b** Heatmap from the set of increased and decreased mRNAs in synaptoneurosomes of *Fus^{ΔNLS/+}* at 6-months compared to *Fus^{+/+}*. mRNAs are on the rows and the different samples on the columns. The color scale indicates the log2FC between the CPM of each sample and mean CPM of the corresponding *Fus^{+/+}* samples at each time point [sample logCPM – mean (logCPM of *Fus^{+/+}* samples)]. **c** Volcano plots showing the log2 fold change of each mRNA and the corresponding minus log10 (FDR) of the differential gene expression analysis comparing total cortex and synaptoneurosomes from *Fus^{ΔNLS/+}* to *Fus^{+/+}* at 1 month (left panel) and 6 months of age (right panel). The horizontal line marks the significance threshold of 0.05. Significantly decreased mRNAs are highlighted in green, increased mRNAs in purple and all FUS targets identified in the CLIP-seq data in blue. **d** Euler diagram of the sets of significantly increased and decreased mRNAs (synaptoneurosomes of *Fus^{ΔNLS/+}* vs. *Fus^{+/+}* at 6 months of age) and the synaptoneurosomes FUS targets identified by CLIP-seq. The *p*-values indicated the enrichment of FUS targets among the increased/decreased mRNAs (one-sided Fisher's exact test).

that FUS is both directly and indirectly influencing mRNA stability and that cytoplasmic accumulation of FUS leads to altered stability of specific mRNAs.

FUS targets with changed stability at 12 h after actinomycin D treatment significantly correlated with the mRNAs altered in synaptoneurosomes from 6-month-old *Fus^{ΔNLS/+}* mice. When we combined the results from the stability assay with the SNS RNA-seq data at 6 months, we found that FUS targets are enriched in the set of mRNAs that were altered in stability and/or were accumulated at 6 months ($p = 8.8e-07$, one-sided Fisher's exact test). However, the set of mRNAs with decreased stability and/or decreased abundance are not enriched for FUS targets ($p = 1$, one-sided Fisher's exact test, Supplementary Fig. 9e). Overall, our analysis suggested that accumulated levels of a subset of synaptic RNA targets in 6-month-old *Fus^{ΔNLS/+}* mice may be due to altered stability caused by increased FUS binding within their exons (Fig. 5g, h). Altogether, FUS mislocalization leads to alterations in the synaptic RNA profile, which may affect synaptic signaling and plasticity. Synaptic RNA alterations represent one of the earliest events in disease pathogenesis, suggesting that ALS-FUS is, at least partly, a synaptopathy.

## Discussion

In this study, we identified for the first-time synaptic RNA targets of FUS combining cortical synaptoneurosome preparations with CLIP-seq. We also assessed FUS localization at the synaptic site using a combination of super-resolution microscopy approaches. Altogether, our results point to a critical role for FUS at the synapse and indicate that increased synaptic FUS localization in ALS-FUS mice triggers early alterations of synaptic RNA content and misregulation of the GABAergic network. These early synaptic changes correlate with the behavioral dysfunctions that these mice develop[45].

CLIP-seq on synaptoneurosome preparations from mouse cortex demonstrated that FUS not only binds nuclear RNAs, but also those that are localized at both pre- and postsynapses. Moreover, we identified that FUS binds RNAs encoding GABA receptor subunits (*Gabra1, Gabrb3, Gabbr1, Gabbr2*) and glutamatergic receptors (*Gria2, Gria3, Grin1*), previously known to be localized at dendritic neuropils[77]. The same FUS RNA targets (*Gabra1, Grin1, Gria2, Gria3*) were found increased at the synapses of *Fus^{ΔNLS/+}* mice. This is in agreement with an independent study[45] that also reported increased levels of FUS RNA

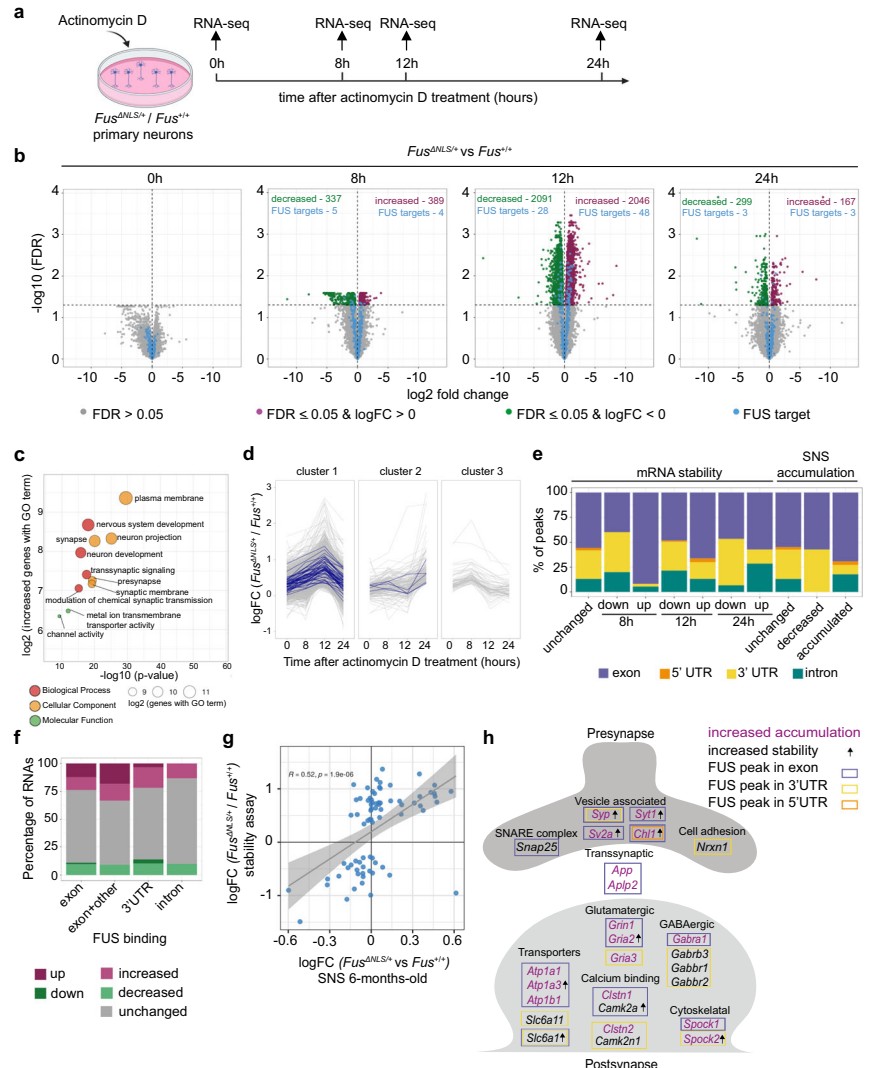

**Fig. 5 Cytoplasmic FUS accumulation in *Fus*^ΔNLS/+^ neurons leads to increased stability of synaptic mRNAs. a** Workflow of the RNA stability experiment. **b** Volcano plots showing the log2 fold change of each mRNA and the corresponding −log10 (FDR) of the differential gene expression analysis comparing *Fus*^ΔNLS/+^ to *Fus*^+/+^ samples at each time point with significance threshold 0.05 (horizontal line). Significantly decreased mRNAs are highlighted in green, increased in purple and all synaptic FUS targets identified in the CLIP-seq data in blue. **c** Selected GO terms enriched in the set of significantly increased mRNAs at 12 h. One-sided hypergeometric test, *p*-value unadjusted for multiple comparisons. **d** Hierarchical clustering of the log fold changes of the set of increased mRNAs to three groups. The *x*-axis shows the time points and the y-axis the logFC (*Fus*^ΔNLS/+^/*Fus*^+/+^); synaptic FUS targets are highlighted in blue. **e** Bar plot of different sets of synaptic FUS targets and the location of synapse-specific peaks in these genes. FUS targets with no stability change in *Fus*^ΔNLS/+^ compared to *Fus*^+/+^ (unchanged) and the sets of significantly changed mRNAs (up or down) at 8, 12, and 24 h after treatment. The three rightmost bars comprise the FUS targets with no change in synaptoneurosomes (SNS) at 6-months (unchanged), accumulated and decreased targets. **f** Bar plot of FUS targets grouped by binding location and the percentage of targets with a specific RNA level change. Targets with the majority of peaks in exons (exon), equal number of peaks in exons and UTR/intron (exon + other), majority of peaks in 3′UTR (3′UTR) and majority of peaks in introns (intron). Accumulated RNAs in 6-month-old SNS (up), RNAs with increases stability (increased), unchanged RNA levels (unchanged), less abundant RNAs in 6-month-old SNS (down) and less stable RNAs (decreased). **g** Log2 fold change (*Fus*^ΔNLS/+^/*Fus*^+/+^) of the stability experiment at 12 h (*y*-axis) and SNS of 6-month-old mice (*x*-axis) for all synaptic FUS targets with significant changes in stability at 12 h (blue points, *n* = 76 mRNAs). *R* = Pearson correlation coefficient, *p* = p-value (two-sided Pearson correlation test), linear fit (gray line) and 95% confidence interval (gray region). **h** Summary of the observed changes for selected synaptic FUS targets. RNAs that were accumulated in synaptoneurosomes of 6-month-old *Fus*^ΔNLS/+^ mice are highlighted in magenta; RNAs with increased stability at any time point are indicated by a small arrow; the colored boxes indicate the location of the synaptoneurosome-specific FUS peaks.

targets, including *Gabra1* and *Nrxn1*, in synaptosomal fractions of 5-month-old *Fus*^ΔNLS/+^ mice.

Our image analysis showed alterations of inhibitory synapses at 1- and 6-month-old *Fus*^ΔNLS/+^ mice. We explored GABA_AR density and found changes in α3-containing GABA_AR. GABA_Aα3 is expressed at the postsynaptic site of monoaminergic synapses[78], which have been shown to be involved in fear and

anxiety behavior, and mutations in the *Gabra3* subunit resulted in an absence of inhibition behavior[79–81]. Changes in GABA_Aα3 and not GABA_Aα1-containing receptor suggested that only mono-aminergic neurons were affected in the *Fus*^ΔNLS/+^ mouse model. These results are well aligned with a contemporaneous study[45], which showed specific behavioral changes that can be linked to monoaminergic networks. Interestingly, at 1 month of age,

$Fus^{\Delta NLS/+}$ mice showed an increase of NMDAR and a decrease in GABA$_A$α3. These results suggested a role for FUS during synaptogenesis in regulating postsynaptic receptor composition, as previously suggested[23,28,82]. In 1-month-old $Fus^{\Delta NLS/+}$ mice, NMDARs were enriched at the extrasynaptic sites, which, together with the decrease in GABA$_A$α3, suggested a hyperexcitability profile during development, which may result in abnormal network connection. $Fus^{\Delta NLS/+}$ mice at 6 months of age showed higher density of presynaptic inhibitory boutons, pointing toward a compensatory mechanism at the GABAergic synapses to overcome the hyperexcitability profile observed during development. We also observed an increase in GABA$_A$α3 cluster size and density in 6-month-old $Fus^{\Delta NLS/+}$ mice, which was surprisingly not accompanied by an increase in Gephyrin, a postsynaptic protein responsible for anchoring GABAR at the postsynaptic site[83,84] at a ratio 1:1[85]. Collectively, these findings suggest that inhibitory $Fus^{\Delta NLS/+}$ synapses were unstable at 6 months of age with an excess of GABAR poorly anchored at the postsynaptic site, which could lead to malfunction of the inhibitory network.

Using the $Fus^{\Delta NLS/+}$ mouse model, we found that synaptic accumulation of mutant FUS altered the synaptic RNA content as early as 6 months of age. Decreased mRNAs encoded proteins linked to metabolism and were depleted from FUS binding, suggesting that these alterations were indirect consequences of cytoplasmic and/or synaptic FUS accumulation. In contrast, mRNAs with increased accumulation at the synaptic sites encoded proteins important for synaptic function and were enriched for synaptic FUS targets. This suggested that increased synaptic FUS localization directly augmented the accumulation of these synaptic targets. While the precise mechanism of how FUS regulates these targets will be the focus of future investigations, our analysis shows that a subset of synaptic RNA targets is stabilized by FUS binding on multiple exonic sites. Indeed, this is suggested by the significant enrichment of exonic binding on mRNAs that are either increased in stability or accumulated in $Fus^{\Delta NLS/+}$ mice and reversely, by the enrichment of accumulated or stabilized mRNAs among synaptic FUS targets with exonic binding.

In contrast, mRNAs with 3′UTR FUS binding remained largely unchanged in our experiments. Our CLIP-seq from synapto-neurosomes showed that FUS binds to the last exon of the long 3′ UTR-containing isoform of Gabra1 (Supplementary Fig. 8) indicating that FUS may be directly involved in regulating the protein translation of Gabra1 at the synapses. In fact, all synaptic FUS targets with 3′UTR binding, including those encoding the GABAR subunits, NRXN1 and CAMK2N1 are strong candidates for local translation regulation directly via synaptic FUS binding at the synapse. The novel binding motif that we identified specifically among the synaptic 3′UTR targets implies a distinct mechanism, potentially via binding on secondary structures via the RRM of FUS, in line with recent structural studies[42,43].

Among the altered synaptic FUS targets is App, encoding the amyloid precursor protein (APP), a transmembrane protein with crucial roles in synaptic function and stability[86–88]. Importantly, APP is causally linked with Alzheimer's disease and our observations suggest a previously unidentified link between synaptic FUS accumulation and APP misregulation. The recent observation that APP contributes to the regulation of inhibitory synapses[89,90] reinforces this link.

Dissecting the exact molecular underpinnings of synaptic FUS-mediated regulation of mRNA stability and local translation is now an extremely important next step. Our work indicates that early synaptopathy triggered by synaptic FUS accumulation, prior to aggregation, leads to ALS-FUS and understanding the underlying molecular events will be key for devising early and effective therapeutic interventions.

## Methods

**Experimental models.** Mice housing and breeding were in accordance with the Swiss Animal Welfare Law and in compliance with the regulations of the Cantonal Veterinary Office, Zurich. We used 1- to 6-month-old C57/Bl6 mice or $Fus^{+/+}/Fus^{\Delta NLS/+}$ mice with genetic background (C57/Bl6). Wild type and heterozygous $Fus^{\Delta NLS/+}$ mice with genetic background (C57/Bl6)[44] were bred and housed in the animal facility of the University of Zurich.

**Immunofluorescence staining for brain sections.** Mice were anesthetized by $CO_2$ inhalation before perfusion with PBS containing 4% paraformaldehyde and 4% sucrose. Brains were harvested and post-fixed overnight in the same fixative and then stored at 4 °C in PBS containing 30% sucrose. Sixty μm-thick coronal sections were cut on a cryostat and processed for free-floating immunofluorescence staining. Brain sections were incubated with the indicated primary antibodies for 48 h at 4 °C followed by secondary antibodies (Invitrogen) for 24 h at 4 °C. The antibodies were diluted in 1X Tris Buffer Saline solution containing 10% donkey serum, 3% BSA, and 0.25% Triton-X100. Sections were then mounted on slides with Prolong Diamond (Life Technologies) before confocal microscopy. The list of antibodies is provided in Supplementary data 9.

**STED super-resolution imaging and analysis.** Super-resolution STED (Stimulated emission depletion microscopy) images of FUS and synaptic markers were acquired on a Leica SP8 3D, 3-color gated STED laser scanning confocal microscope. Images were acquired in the retrospenial cortical area in the layer 5 and in the molecular layer of the hippocampal CA1 area. A 775-nm depletion laser was used to deplete both 647 and 594 dyes. The powers used for depletion lasers, the excitation laser parameters, and the gating parameters necessary to obtain STED resolution were assessed for each marker and kept constant. In all, 1-μm-thick Z-stacks of 1024 × 1024-pixel images at 40 nm step size were acquired at 1800 kHz bidirectional scan rate with a line averaging of 32 and 3 frame accumulation, using a ×100 (1.45) objective with a digital zoom factor of 7.5, yielding 15.15 nm pixels resolution.

STED microscopy data were quantified from at least 2 image stacks acquired from 2 $Fus^{+/+}$ adult mice. The STED images were deconvolved using Huygens Professional software (Scientific Volume Imaging). Images were subsequently analyzed using Imaris software. Volumes for each marker were generated using smooth surfaces with details set up at 0.01 m. The diameter of the largest sphere was set up at 1 μm. Threshold background subtraction methods were used to create the surface, and the threshold was calculated for each marker and kept constant. Surfaces were then filtered by setting up the number of voxels >10 and <2000 pixels. Closest neighbor distance was calculated using integrated distance transformation tool in Imaris. Distances were then organized and statistically analyzed using mean comparison and t-test comparison. Distances >200 nm were removed from the analysis, and average distance were analyzed.

**Neuronal primary cultures.** Primary neuronal cell cultures were prepared from postnatal (P0) pups. Briefly, hippocampus and cortex were isolated. Hippocampi were treated with trypsin (0.5% w/v) in HBSS-Glucose (D-Glucose, 0.65 mg/ml) and triturated with glass pipettes to dissociate tissue in Neurobasal medium (NB) supplemented with glutamine (2 mM), 2% B27, 2.5% Horse Serum, 100 U penicillin-streptomycin, and D-Glucose (0.65 mg/ml). Hippocampal cells were then plated onto poly-D-lysine coated 18 × 18 mm coverslips (Carl Zeiss$^{TM}$ - 10474379) at 6 × 10$^4$ cells/cm² for imaging, and for biochemistry at high density (8 × 10$^4$ cells/cm²). Cells were subsequently cultured in supplemented Neurobasal (NB) medium at 37 °C under 5% $CO_2$, one-half of the medium changed every 5 days, and used after 15 days in vitro (DIV). Cortex were dissociated and plated similarly to hippocampal cells in NB supplemented with 2% B27, 5% horse serum, 1% N2, 1% glutamax, 100 U penicillin-streptomycin and D-Glucose (0.65 mg/ml).

**Stability assay.** Mixed hippocampal and cortical neuronal cultures were prepared from mouse embryos (E17).

Briefly, hippocampus and cortex were isolated from each embryo. The tissues were digested with trypsin (0,5% w/v supplemented with 4% w/v D-Glucose) and cells were plated on poly-D-lysine coated 6-well-plates: primary neurons derived from one embryo were seeded on four wells, one for each Actinomycin D treatment time point. Cells were plated in Neurobasal medium (NB) supplemented with glutamine (2 mM), 2% B27, 2% N2, 100 U penicillin-streptomycin and 4% w/v D-glucose. At DIV 12, cultures were treated with 10 μg/mL Actinomycin D for 24 h,12 h, 8 h, or 0 h (untreated). Each time point consists of six replicates. Primary neuronal cultures from each embryo were collected at the same time and total RNA was isolated using RNeasy Plus Mini Kit (Cat No. 79254, Qiagen). Single-end poly (A) containing cDNA libraries were prepared and were sequenced on Hiseq2500.

**Direct Stochastic Optical Reconstruction Microscopy (dSTORM).** Super-resolution images were acquired on a Leica SR Ground State Depletion 3D / 3 color TIRFM microscope with an Andor iXon Ultra 897 EMCCD camera (Andor Technology PLC). DIV15-18 mouse primary neurons were fixed for 20 min in 4% PFA – 4% sucrose in PBS. Primary antibodies (Supplementary data 9) were incubated overnight at 4% in PBS containing 10% donkey serum, 3% BSA, and

0.25% Triton X-100. Secondary antibodies (Invitrogen) were incubated at RT for 3 h in the same buffer. After three washes in PBS, the cells were re-fixed with 4%PFA for 5 min. The coverslips were then washed over a period of 2 days at 4 °C in PBS to remove non-specific binding of the secondary antibodies. Coverslips were mounted temporarily in an oxygen scavenger buffer (200 mM phosphate buffer, 40% glucose, 1 M cysteamine hydrochloride (M6500 Sigma), 0.5 mg/mL Glucose-oxydase, 40ug/mL Catalase) to limit oxidation of the fluorophores during image acquisition. The areas of capture were blindly selected by direct observation in DIC. Images were acquired using a ×160 (NA 1.43) objective in the TIRF mode North direction with a penetration of 200 nm. Far red channels (Alexa 647 or 660) were acquired using a 642 nm laser. Red channels (Alexa 568 or 555) were acquired using a 532 nm laser. Green channel (Alexa 488) was acquired using 488 nm laser. Images were acquired in 2D. The irradiation intensity was adjusted until the single molecule detection reached a frame correlation <0.25. Detection particle threshold was defined between 20-60 depending on the marker and adjusted to obtain a number of events per frame between 0 and 25. The exposure was maintained at 7.07 ms and the EM gain was set at 300. The power of depletion and acquisition was defined per marker and kept constant during acquisition. The number of particles collected were maintained constant per markers and between experiments. At least three independent cultures or coverslips were imaged per marker.

**Super-resolution image processing and analysis**. Raw GSD images were processed using a custom-made macro in Fiji to remove background by subtraction of a running median of frames (300 renewed every 300 frames) and subtracting the previously processed image once background was removed[91]. A blur (0.7-pixel radius) per slice prior to median subtraction was applied to reduce the noise further. These images were then processed using Thunderstorm plugin in Imagej. Image filtering was performed using Wavelet filter (B-spline, order 3/scale2.0). The molecules were localized using centroid of connected components, and the peak intensity threshold was determined per marker/dye to maintain an XY uncertainty <50. Sub-pixel localization of molecules was performed using PSF elliptical gaussian and least squared fitting methods with a fitting radius of 5 pixels and initial sigma of 1.6 pixels. Images were analyzed using Bitplane Imaris software v.9.3.0 (Andor Technology PLC). Volumes for each marker were generated using smooth surfaces with details set up at 0.005. The diameter of the largest sphere was set up at 1 μm. A threshold background subtraction method was used to create the surface and threshold was calculated and applied to all the images of the same experiment. Surfaces were then filtered by setting up the area between 0.01 and 1 μm². The closest neighbor distance was processed using the integrated distance transformation tool in Imaris. Distances were then organized and statistically analyzed using median comparison and ANOVA and Fisher's Least Significant Difference (LSD) test. Distances >100 nm were removed from the analysis, and average distance were analyzed.

**Preparation of synaptoneurosomes from mouse brain tissues**. Synaptoneurosomes were prepared based on previously published protocols[92,93] with slight modifications. The freshly harvested cortex tissue was homogenized using dounce homogenizer for 12 strokes at 4 °C in buffer (10%w/v) containing pH 7.4, 10 mM 4-(2 hydroxyethyl)-1-piperazineethanesulfonic acid (HEPES; Biosolve 08042359), 0.35 M Sucrose, 1 mM ethylenediaminetetraacetic acid (EDTA; VWR 0105), 0.25 mM dithiothreitol (Thermo Fisher Scientific R0861), 30 U/ml RNAse inhibitor (Life Technologies N8080119), and complete- EDTA free protease inhibitor cocktail (Roche 11836170001, PhosSTOP (Roche 04906845001). In all, 200 μl of the total cortex homogenate were saved for RNA extraction or western blot analysis. The remaining homogenate was spun at 1000×g, 15 min at 4 °C to remove the nuclear and cell debris. The supernatant was sequentially passed through three 100 μm nylon net filters (Millipore NY1H02500), followed by one 5-μm filter (Millipore SMWP013000). The filtrate was resuspended in three volumes of SNS buffer without sucrose and spun at 2000×g, 15 min at 4 °C to collect the pellet containing synaptoneurosomes. The synaptoneurosome pellets were resuspended in RIPA buffer for western blot or in QIAzol reagent for RNA extraction or submitted as a pellet for sample processing for proteomics.

**Cross-linking immunoprecipitation and high-throughput sequencing**. Total lysate and synaptoneurosomes isolated from cortex tissue of 1-month-old C57Bl/6 mice were UV crosslinked (100 mJ/cm² for 2 cycles) using UV Stratalinker 2400 (Stratagene) and stored at −80 °C until use. For the total sample, cortex tissue was dissociated using a cell strainer of pore size 100 μm before crosslinking. We used cortex from 200 mice to prepare SNS and two mice for the total cortex sample. We used a mouse monoclonal antibody specific for the C-terminus of FUS (Santa Cruz; sc-47711) (Supplementary data 9) to pull down FUS-associated RNAs using magnetic beads. After immunoprecipitation, FUS-RNA complexes were treated with MNAse in mild conditions and the 5′ end of RNAs were radiolabeled with P32-gamma ATP. Samples run on SDS-gel (10% Bis Tris) were transferred to nitrocellulose membrane and visualized using FLA phosphorimager. RNAs corresponding to FUS-RNA complexes were purified from the nitrocellulose membrane. Strand-specific paired-end CLIP libraries were prepared by using The SMARTer Stranded Total RNA-Seq Kit - Pico Input Mammalian (Clontech Laboratories, Inc., A Takara Bio Company, California, USA) without the ribosomal

depletion step followed by PCR amplification for 15 cycles. The libraries were sequenced on HiSeq 2500.

**Bioinformatic analysis of CLIP-seq data and identification of FUS targets**. Low quality reads were filtered and adapter sequences were removed with Trim Galore! (Krueger, F., TrimGalore. Retrieved February 24, 2010, from https://github.com/FelixKrueger/TrimGalore). Reads were aligned to the mouse reference genome (build GRCm38) using STAR version 2.4.2a[94] and Ensembl gene annotations (version 90). We allowed a maximum of two mismatches per read (--outFilterMismatchNmax 2) and removed all multimapping reads (--outFilterMultimapNmax 1). PCR duplicates were removed with Picard tools version 2.18.4 ("Picard Toolkit." 2019. Broad Institute, GitHub Repository. http://broadinstitute.github.io/picard/; Broad Institute). Peaks were called separately on each sample with CLIPper[55] using default parameters.

To identify regions that are specifically bound by FUS in the SNS sample but not the total cortex sample, we filtered the peaks based on an MA plot. For each peak, we counted the number of overlapping reads in the SNS ($x$) and total cortex samples ($y$). M (log2 fold change) and A (average log2 counts) were calculated as follows:

$$M = \log 2[(x + o)/(\text{lib.size}\_x + o)] - \log 2[(y + o)/(\text{lib.size}\_y + o)] \quad (1)$$

$$A = [\log 2(x + o) + \log 2(y + o)]/2 \quad (2)$$

where $o = 1$ is an offset to prevent a division by 0 and lib.size_x and lib.size_y is the effective library size of the two samples: the library size (number of reads mapping to the peaks) multiplied by the normalization factor obtained from "calcNormFactors" using the trimmed mean of M-values[95] method. The M and A values of all CLIPper peaks identified in the SNS sample were plotted against each other (x-axis A, y-axis M). The plot was not centered at a log2FC of 0. Therefore, we fitted a LOESS (locally estimated scatterplot smoothing) curve for normalization (loess(formula=M~A, span=1/4, family = "symmetric", degree=1, iterations=4)). We computed the predicted M values (fitted) for each A value and adjusted the M values by the fit (adjusted M = M - fitted M). After adjustment, the fitted LOESS line crosses the y-axis at 0 with slope = 0 in the adjusted MA-plot.

For ranking purposes, we computed p-values for each peak with the Bioconductor edgeR package[95]. We computed the common dispersion of the peaks at the center of the main point cloud (-3 < y < 1 in raw MA-plot) and not the tagwise dispersion because we are lacking replicate information. Peak specific offsets were computed as log (lib.size*norm.factors) where norm.factors are the normalization factors. The fitted M-values were subtracted from the peak specific offsets to use the adjustments from the LOESS fit for the statistical inference. We fit a negative binomial generalized linear model to the peak specific read counts using the adjusted offsets. We want to test for differential read counts between the synaptoneurosome and total cortex sample (~group). A likelihood ratio test[96] was run on each peak to test for synaptoneurosome versus total cortex differences.

We compared the sets of peaks obtained from different p-value cutoffs (Supplementary Fig. 2g) and choose the most stringed cutoff of 1e-5 because it showed the strongest depletion of intronic peaks and strongest enrichment of exonic and 3′UTR peaks. CLIPper annotated each peak to a gene and we manually inspected the assigned genes and removed wrong assignments caused by overlapping gene annotations.

Total cortex-specific peaks (regions that are exclusively bound in the total cortex sample but not the SNS sample) were computed with the same approach: The M values were computed as

$$M = \log 2((y + o)/(\text{lib.size}\_y + o)) - \log 2((x + o)/(\text{lib.size}\_x + o)) \quad (3)$$

and we used a p-value cutoff of 0.0029825 because that resulted in an identical number of SNS-specific peaks.

For the over representation analysis (ORA) we applied the "goana" function from the limma R package using the gene length as covariate[97]. As background set, we used all genes with a cpm of at least 1 in all RNA-seq samples of synaptoneurosomes from 1-month-old mice.

RNA motifs of length 2–8 were predicted with HOMER[60]. To help with the motif finding, we decided to use input sequences of equal length because the lengths of the predicted peaks varied a lot. We define the peak center as the median position with maximum read coverage. Then, we centered a window of size 41 on the peak center of each selected peak and extracted the genomic sequence. We generated background sequences for each set of target sequences. A background set consists of 200,000 sequences of length 41 from random locations with the same annotation as the corresponding target set (intron, exon, 3′UTR or 5′ UTR). All background sequences are from regions without any read coverage in the corresponding CLIP-seq sample to ensure that the background sequences are not bound by FUS.

**RNA extraction and high-throughput sequencing (RNA-seq)**. Cortex tissue was isolated from 1 and/or 6-month-old $Fus^{\Delta NLS/+}$ and $Fus^{+/+}$ mice. Paired total cortex (200 μl) and SNS sample was obtained from a single mouse per condition. Briefly, frozen total and SNS samples were mixed with QIAzol reagent following the manufacturer's recommendations and incubated at RT for 5 min. Two hundred microliters of chloroform were added to the samples and mixed for 15 s and then

centrifuged for 15 min (12,000 × g, 4 °C). To the upper aqueous phase collected, five hundred microliters of isopropanol and 0.8 μl of glycogen was added and incubated at RT for 15 min. The samples were centrifuged at 10,000 rpm for 10 min. After centrifugation at 12,000 × g for 15 min, the isopropanol was removed and the pellet was washed with 1 ml of 70% ethanol and samples were centrifuged for 5 min at 7500 × g. Ethanol was discarded and the RNA pellet was air-dried and dissolved in nuclease free water and further purified using the RNeasy Mini Kit including the DNAse I digestion step. The concentration and the RIN values were determined by Bioanalyzer. In all, 150 ng of total RNA were used for Poly A library preparation. Strand-specific cDNA libraries were prepared and sequenced on Illumina NovaSeq6000 platform (2x150 bp, paired end) from Eurofins Genomics, Konstanz, Germany.

**cDNA synthesis and quantitative real-time PCR**. RNA was isolated from synaptoneurosomes (as described before) and paired total cortical lysate from three independent 1-month-old C57/Bl6 mice using QIAzol reagent. Total RNA was reverse transcribed using Superscript III kit (Invitrogen). For qRT-PCR, 2x SYBR master mix (Thermoscientific) were used and the reaction was run in Thermo-cycler (Applied Biosystems ViiA 7) following the manufacturer's instructions. The primers used for qRT-PCR were prepared using NCBI Primer-Blast tool (Supplementary data 10)

**SDS-PAGE and western blotting**. Protein concentrations were determined using the Pierce BCA Protein Assay (Thermo Fisher Scientific) prior to SDS-PAGE. In all, 20 μg for total protein were used for western blots. The samples were resuspended in 1X SDS loading buffer with 1X final sample reducing reagent and boiled at 95 °C, 10 mins. Samples were separated by Bolt 4-12% Bis-Tris pre-cast gels and transferred onto nitrocellulose membranes using iBlot® transfer NC stacks with iBlot Dry Blotting system (Invitrogen). Membranes were blocked with buffer containing 0.05% v/v Tween-20 (Sigma P1379) prepared in PBS (PBST) with 5% w/v non-fat skimmed powdered milk and probed with primary antibodies (Supplementary data 9) overnight at 4 °C in PBST with 1% w/v milk. Following three washes with PBST, membranes were incubated with secondary HRP-conjugated goat anti mouse or rabbit AffiniPure IgG antibodies (1:5000, 1:10,000, respectively) (Jackson ImmunoResearch 115-035-146 and 111-035-144, respectively) in PBST with 1% w/v milk, for 1.5 h at RT. Membranes were washed with PBST, and the bands were visualized using Amersham Imager 600RGB (GE Healthcare Life Sciences 29083467).

**Bioinformatic analysis of RNA-seq data**. The preprocessing, gene quantification and differential gene expression analysis was performed with the ARMOR workflow[98]. In brief, reads were quality filtered and adapters were removed with Trim Galore! (Krueger, F., TrimGalore. Retrieved February 24, 2010, from https://github.com/FelixKrueger/TrimGalore). After preprocessing, the number of reads per sample ranged from 25 million to 50 million with a median of 35 million. For visualization purposes, reads were mapped to the mouse reference genome GRCm38 with STAR version 2.4.2a[94] and default parameters using Ensembl gene annotations (version 90). BAM files were converted to BigWig files with bedtools[99]. Transcript abundance estimates were computed with Salmon version 0.10.2[100] and summarized to gene level with the tximeta R package[101]. All downstream analyses were performed in R and the edgeR package[95] was used for differential gene expression analysis. We filtered the lowly expressed genes and kept all genes with a CPM of at least 10/median_library_size*1e6 in 4 replicates (the size of the smallest group, here the nuclear samples). Additionally, each kept gene is required to have at least 15 counts across all samples. The filtered set of genes was used for the PCA plot and differential gene expression analysis.

For the over representation analysis, we applied the "goana" function from the limma R package using the gene length as covariate[97]. As background set, we used all genes with a cpm of at least 1 in all $Fus^{+/+}$ SNS replicates at 6 months. One-sided Fisher's exact test was used to test for enrichment of FUS targets among the set of increased or decreased mRNAs.

**Bioinformatic analysis of stability assay**. The stability assay was analyzed with the same ARMOR workflow and reference annotation as for the other RNA-seq dataset including the differential gene expression analysis. All analyses were performed in R. Differentially expressed genes in $Fus^{ΔNLS/+}$ compared to $Fus^{+/+}$ were determined at each time point with an FDR threshold of 0.05.

We focused our analyses on the protein coding genes and lincRNAs. Principal components were computed on the log cpm values of the 1000 genes with the highest variance. Over representation analysis was performed with the "goana" function from the limma R package. As a background universe, we used all genes with >1 cpm in all samples at time point 0 h. The top and non-redundant GO terms were selected for visualization.

For all further analyses, the genes were subset to the ones expressed in WT SNS at 1 month (read count > 10 in at least two thirds of the replicates). Differentially expressed genes from any of the time points were divided into genes with decreased (logFC < 0) or increased (logFC > 0) expression in $Fus^{ΔNLS/+}$. Agglomerative hierarchical clustering was performed on the two sets of genes using the "hclust" and "cutree" R functions. The Pearson correlation coefficient of the log fold-changes ($Fus^{ΔNLS/+}/Fus^{+/+}$) from the four time points was used as a distance measure [½(1-R)] between pairs of genes. We used average linkage clustering and cut the two trees into three groups each.

For all genes in the increased and decreased set, we computed the length of the longest transcript, CDS, 3′UTR and 5′UTR. We performed a one-sided unpaired t-test to test for differences in the mean of feature lengths in the increased and decreased gene sets. The length distribution in the different clusters per gene set was compared with ANOVA and posthoc Tukey's HSD.

To test for association of FUS binding with the direction of RNA level change in the stability assay and at 6 months in SNS, we performed Pearson's Chi-squared tests (chisq.test in R). At 8 h post treatment, we tested unchanged and up; at 12 h we tested unchanged, down and up; for SNS we tested unchanged and accumulated. All other target sets were excluded because they had expected frequencies < 5.

An enrichment of FUS targets among the set of increased and/or accumulated RNAs and the decreased and/or less abundant RNAs was tested with one-sided Fisher's exact test.

**Sample preparation for proteomics**. Samples were prepared by using a commercial iST Kit (PreOmics, Germany) with an updated version of the protocol. Briefly, 25 ug of the samples were solubilized in 'Lyse' buffer, boiled at 95 °C for 10 min and processed with High Intensity Focused Ultrasound (HIFU) for 30 s setting the ultrasonic amplitude to 85%. Then the samples were transferred to the cartridge and digested by adding 50 μl of the 'Digest' solution. After 60 min of incubation at 37 °C the digestion was stopped with 100 μl of Stop solution. The solutions in the cartridge were removed by centrifugation at 3800×g, while the peptides were retained by the iST-filter. Finally, the peptides were washed, eluted, dried and re-solubilized in 20 μl 'LC-Load' buffer for MS- Analysis.

**Liquid chromatography-mass spectrometry analysis**. Mass spectrometry analysis of SNS samples was performed on a Q Exactive HF-X mass spectrometer (Thermo Scientific), while lysates of total cortex were analyzed on an Orbitrap Fusion Lumos. Both instruments were equipped with a Digital PicoView source (New Objective) and coupled to a M-Class UPLC (Waters). Solvent composition at the two channels was 0.1% formic acid for channel A and 0.1% formic acid, 99.9% acetonitrile for channel B. For each sample 1 μL of peptides were loaded on a commercial MZ Symmetry C18 Trap Column (100 Å, 5 μm, 180 μm × 20 mm, Waters) followed by nanoEase MZ C18 HSS T3 Column (100 Å, 1.8 μm, 75 μm × 250 mm, Waters). The peptides were eluted at a flow rate of 300 nL/min by a gradient from 8 to 27% B in 85 min, 35% B in 5 min and 80% B in 1 min. Samples were acquired in a randomized order. The mass spectrometers were operated in data-dependent mode (DDA). For Q Exactive HF-X analyses, DDA was performed acquiring a full-scan MS spectrum ($350 − 1400$ m/z) at a resolution of 120,000 at 200 m/z after accumulation to a target value of 3,000,000, followed by HCD (higher-energy collision dissociation) fragmentation on the twenty most intense signals per cycle. HCD spectra were acquired at a resolution of 15,000 using a normalized collision energy of 25 and a maximum injection time of 22 ms. The automatic gain control (AGC) was set to 100,000 ions. Only precursors with intensity above 110,000 were selected for MS/MS. For Orbitrap Fusion Lumos analyses, all precursor signals were recorded in the Orbitrap using quadrupole transmission in the mass range of 300–1500 m/z. Spectra were recorded with a resolution of 120,000 at 200 m/z, a target value of 500,000 and the maximum cycle time was set to 3 s. Data- dependent MS/MS were recorded in the linear ion trap using quadrupole isolation with a window of 0.8 Da and HCD fragmentation with 35% fragmentation energy. The ion trap was operated in rapid scan mode with a target value of 10,000 and a maximum injection time of 50 ms. Only precursors with intensity above 5000 were selected for MS/MS. For both instrument, charge state screening was enabled. Singly, unassigned, and charge states higher than seven were rejected. Precursor masses previously selected for MS/MS measurement were excluded from further selection for 30 s, and the exclusion window was set at 10 ppm. The samples were acquired using internal lock mass calibration on m/z 371.1012 and 445.1200. The mass spectrometry proteomics data were handled using the local laboratory information management system (LIMS)[102].

**Protein identification and label free protein quantification**. The acquired raw MS data were processed by MaxQuant (version 1.6.2.3), followed by protein identification using the integrated Andromeda search engine[103]. Spectra were searched against a Swissprot Mus musculus (Mouse) reference proteome (taxonomy 10090, version from 2019-07-09), concatenated to its reversed decoyed fasta database and common protein contaminants. Carbamidomethylation of cysteine was set as fixed modification, while methionine oxidation and N-terminal protein acetylation were set as variable. Enzyme specificity was set to trypsin/P allowing a minimal peptide length of seven amino acids and a maximum of two missed-cleavages. MaxQuant Orbitrap default search settings were used. The maximum false discovery rate (FDR) was set to 0.01 for peptides and 0.05 for proteins. Label free quantification was enabled and a 2-min window for match between runs was applied. In the MaxQuant experimental design template, each file is kept separate in the experimental design to obtain individual quantitative values.

**Bioinformatics data analysis of mass spectrometry data**. Protein fold changes were computed based on Intensity values reported in the proteinGroups.txt file. A set of functions implemented in the R package SRMService (http://github.com/protViz/SRMService) was used to filter for proteins with 2 or more peptides allowing for a maximum of 4 missing values, and to normalize the data with a modified robust $z$-score transformation and to compute $p$-values using the $t$-test with pooled variance. If all measurements of a protein are missing in one of the conditions, a pseudo fold change was computed replacing the missing group average by the mean of 10% smallest protein intensities in that condition.

**Transmission electron microscopy**. SNS pellets were prepared from cortical tissue of 1-month-old C57/Bl6 mice as previously mentioned before and submitted to imaging facility at ZMB UZH. Briefly, SNS pellet prepared were resuspended in 2X fixative (5% Glutaraldehyde in 0.2 M Cacodylate buffer) and fixed at RT for 30 mins. Sample was then washed twice with 0.1 M Cacodylate buffer before embedding into 2% Agar Nobile. Post-fixation was performed with 1% Osmium 1 h on ice, washed three times with ddH$_2$O, dehydrated with 70% ethanol for 20 mins, followed by 80% ethanol for 20 mins, 100% for 30 mins, and finally Propylene for 30 mins. Propylene: Epon Araldite at 1:1 were added overnight followed by addition of Epon Araldite for 1 h at RT. Sample was then embedded via 28 h incubation at 60 °C. The resulting block was then cut into 60 nm ultrathin sections using ultramicrotome. Ribbons of sections were then put onto TEM grid and imaged on TEM - FEI CM100 electron microscope (modify).

**Confocal image acquisition and analysis**. Confocal images were acquired on a Leica SP8 Falcon microscope using 63X (NA 1.4) with a zoom power of 3. Images were acquired at a 2048 × 2048 pixel size, yielding to a 30.05-nm/pixel resolution. To quantify the density of synaptic markers, images were acquired in CA1 region in the apical dendrite area, ~50 μm from the soma, at the bifurcation of the apical dendrite of pyramidal cells, using the same parameters for both genotypes. Images were acquired from top to bottom with a Z step size of 500 nm. Images were deconvoluted using Huygens Professional software (Scientific Volume Imaging). Briefly, stacks were analyzed using the built-in particle analysis function in Fiji[104]. The size of the particles was defined according to previously published studies[85,105,106]. To assess the number of clusters, images were thresholded (same threshold per marker and experiment), and a binary mask was generated. A low size threshold of 0.01 μm diameter and high pass threshold of 1 μm diameter was applied. Top and bottom stacks were removed from the analysis to only keep the 40 middle stacks. For the analysis, the number of clusters per 40z stacks was summed and normalized by the volume imaged (75153.8 μm³). The density was normalized by the control group. The densities were compared by t test for 1- and 6-month-old mice. GluN1 synaptic localization was analyzed by counting the number of colocalized GluN1 clusters with Synapsin 1. Colocalization clusters were generated using ImageJ plugin colocalization highlighter. The default parameters were applied to quantify the colocalization. The number of colocalized clusters were quantified using the built-in particle analysis function in Fiji[104].

**Synaptic density and composition imaging and analysis of primary neuronal culture**. Synaptic density and synapse composition was assayed in 22 DIV neuronal cell cultures. Cultures were fixed in cold 4% PFA with 4% sucrose for 20 min at RT. Primary antibodies were incubated overnight at 4 °C (Supplementary data 9). Secondary antibodies (Invitrogen) were incubated for 3 h at RT. Hippocampal primary culture: pyramidal cells were selected based on their morphology and confocal images were acquired on a Leica SP8 Falcon microscope using 63X (NA 1.4) with a zoom power of 3 and analyzed with Fiji software. After deconvolution (huygens professional), images were subsequently thresholded, and subsequent analyses were performed by an investigator blind to cell culture treatment[107].

**Reporting summary**. Further information on research design is available in the Nature Research Reporting Summary linked to this article.

## Data availability

Raw data were deposited at ArrayExpress Archive of Functional Genomics Data (https://www.ebi.ac.uk/arrayexpress/). Accession codes for RNA-seq are E-MTAB-9212 (total cortex and SNS) and E-MTAB-10104 (stability experiment); E-MTAB-9211 for CLIP-seq libraries. The mass spectrometry proteomics data have been deposited to the ProteomeXchange Consortium via the PRIDE partner repository with the dataset identifier PXD024075. Full western blots and quantification data are included in the Source Data file. Source data are provided with this paper.

## Code availability

Custom code is deposited on github: CLIP-seq analysis (https://github.com/khembach/FUS_CLIPseq), RNA-seq analyis (https://github.com/khembach/FUS_RNAseq) and figures (https://github.com/khembach/FUS_paper), and stability assay analysis (https://github.com/khembach/FUS_stability).

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

## Acknowledgements

We gratefully acknowledge the support of the National Centre for Competence in Research (NCCR) RNA & Disease funded by the Swiss National Science Foundation. SS was supported by Swiss Government Excellence Scholarships for Foreign Scholars. The authors would like to thank Prof. Adriano Aguzzi and Dr. Claudia Scheckel for helpful discussions and Dr. Dorothee Dormann for critical comments on the manuscript. We thank Gery Barmettler and Dr. José María Mateos from ZMB UZH for technical help with TEM. We also thank Catharina Aquino, Lucy Poveda, Paolo Nanni from FGCZ for discussions and technical help on CLIP library preparation, RNA-sequencing and proteomics. We created Figs. 1h, 4a, and 5a; Supplementary figs. 1c and 2c using BioRender. com.

## Author contributions

Conceptualization of the study was carried by S.S., K.M.H., and M.P. S.S. performed synaptoneurosome isolation, CLIP-seq sample preparation, and RNA-seq sample preparation. K.M.H. analyzed the data from CLIP-seq and RNA-seq. S.S., K.M.H., M.D.R., and M.P. developed the strategy to analyze the sequencing data. E.T., M.H.P., M.P.B., J. W., and P.S. provided experimental support. L.D. provided the mouse model and input on the study and SM contributed Ribo-Zero sequencing data and feedback on the analysis. P.D.R. performed immunostaining and image analyses including confocal, STED and dSTORM. S.S., K.M.H., E.T., P.D.R., and M.P. wrote and edited the manuscript. M.D.R., P.D.R., and M.P. provided supervision. M.P. directed the entire study. All authors read, edited, and approved the final manuscript.

## Competing interests

The authors declare no competing interests.
