## [Peer Review File · Nature Communications]

REVIEWER COMMENTS

Reviewer #1 (Remarks to the Author):

In this manuscript, Saharana et al. molecularly characterize a mutant FUS knock-in mouse model mirroring the heterozygous genotype of human patients with FUS mutations. This work focuses on extranuclear functions of FUS, particularly at the synapse, and reports FUS enrichment at presynaptic sites based on super-resolution microscopy data. Purifying SNS fractions from mouse brain, the authors use CLIP-seq to identify the synaptic RNA targets of FUS and crossreference these with differentially expressed genes between FUS mutant and WT mice forebrain, revealing abnormalities in expression of genes critical for maintaining synapses. Immunostaining experiments show alterations in levels and distribution of many of the gene products from these proteins, supporting dysregulation of these pathways. While FUS has been implicated in the transport and translation of RNAs in neuronal processes, there is relatively little known about its targets or their impact on neuronal physiology. As such, this manuscript provides new information regarding the potential function and impact of extranuclear FUS function that may have broad relevance for FUS-mediated ALS and FTD. The manuscript is also well written and the figures beautifully illustrated. Despite this, however, as it stands the manuscript is largely descriptive, and there is limited tie-in between transcriptomic findings and functional consequences. Moreover, methodological and technical caveats complicate the interpretation of data, as outlined below.

Major comments:

- While intriguing, many of the findings are largely descriptive without a clear connection between synaptic FUS, differentially expressed genes/transcripts, and/or downstream functional effects. The demonstration that synaptically-localized FUS binds RNAs related to excitatory and inhibitory synapses is potentially exciting; however, the impact of such binding is unclear, since few differentially expressed genes are direct FUS targets, and few are verified at the protein level. Further, it remains unclear whether extranuclear FUS, or a relative reduction in nuclear FUS, is responsible for these differences.
- Similarly, there is no effort to tie together the changes in differential RNA expression observed at 1 or 6 mo with the observed changes in excitatory or inhibitory synapses detected in these animals. These phenotypes are observed and described, without investigating direct or indirect connections.
- What is the proportion of FUS within synapses vs. total FUS in the cell? Images in Fig. 1 suggest that the pool of synaptic FUS is minimal, with the signal near background levels. With signal this low, negative controls are a must—these might include FUS KO tissue, for instance—to increase confidence in the staining.
- This manuscript finds very few changes in RNA abundance in total forebrain, while the cosubmitted manuscript using the same mouse model finds hundreds of transcriptomic changes in frontal cortex. What accounts for this large discrepancy?
- While intronic reads are certainly de-enriched, it appears that nearly 20% of the FUS binding sites identified by CLIP in the SNS preparations are in introns, bringing up the possibility of nuclear contamination in the SNS preparations.
- Related to this, with such a small proportion of total FUS in the SNS, how are the read counts in the SNS preps (Fig. 2) just as high as in the total cellular preps?
- Given differences between mouse and human, the authors are encouraged to examine FUS distribution in human tissues as well as mouse, thereby confirming conservation of FUS distribution in humans

Minor comments:

- Line 91: should be “in sporadic FTD”
- Line 161: references are needed for using VGAT and PSD95 as markers of inhibitory and excitatory synapses, respectively

- Line 164: the authors' conclusion that "extranuclear FUS preferentially associates with excitatory synapses" is circular, since they only included FUS clusters within 200 nm from a synaptic marker, and therefore cannot comment on the entire pool of extranuclear FUS
- Line 188: these results are contradictory, but do show that FUS is present in both pre- and post-synaptic regions, although more so in the former.
- Line 299: given the nature of the FUS Δ NLS knock in model and the loss of one allele encoding full-length FUS, wouldn't the authors expect an overall reduction in full-length FUS, regardless of localization?
- Line 353: "instable" should be "unstable"
- Is the distribution of FUS any different within spinal motor neurons, vs. cortical or hippocampal neurons?

Figure comments-

Fig 1:

- a- please mention the use of PNF as neuronal marker in the text
- h- please label pre- and post-synaptic regions, as well as yellow dots

Fig. 2:

- The MA plots in Fig. 2e-f each display a row of points in the upper left corner distinct from the rest of the data, but the significance of these rows is unclear.

Sup Fig. 4:

- f, g- while the heat map displays many potential DEGs, the volcano plots do not show this. The reason for this discrepancy is unclear.

Reviewer #2 (Remarks to the Author):

In this study by Sahadevan et al., the authors aim to gain further knowledge about FUS protein functions at the synapse regulating RNA homeostasis and the implications of ALS associated mutations at the synaptic level.

First, they confirmed previously described synaptic localization of FUS, in both excitatory and inhibitory synapses, with a preference at the presynaptic terminal but also present at the postsynaptic site. Second, they aimed to identify FUS-bound RNAs at the synapses. In order to enrich for synaptic localized RNAs the authors prepared synaptoneurosomes (SNS) upon which Crosslinking Immunoprecipitation (CLIP) was performed. The integrity of the SNS fraction is shown by imaging, western blot enrichment for synaptic proteins, and qRT-PCR for synaptic mRNAs. After SNS-CLIP-seq using a single replicate derived from pooled samples, a filtering is applied to identify synaptic exclusive FUS RNA targets. In the 307 RNAs selected, FUS mostly binds to exons and 3'UTRs in contrast to the preferent binding to introns in the whole tissue sample/whole cell. Within FUS synaptic targets there is an enrichment for those involved in transport, localization, trans-synaptic activity, signalling, receptor binding and transmembrane transporter activity. However, this result might not be surprising since those are terms associated to synaptic activity in a list enriched for synaptic protein encoding RNAs.

To study the implications of ALS mutant FUS in synaptic function, the authors use mice lacking the nuclear localization signal of FUS, which accumulates FUS at the cytoplasm, similar to what happens with ALS-associated FUS mutations. The truncated protein not only accumulates at the cytoplasm, but it is also enriched at the synapses to suggest a potential misregulation of the aforementioned synaptic targets could occur. To explore the

consequences of FUS mislocalization, the authors performed IHC and imaging for markers of different synaptic structures, at 1 and 6 months old. They observe an increase in density of NMDAR and a decrease in density of GABAAalpha3 in post-synaptic sites at 1 month suggesting a synaptic hyperexcitability. Increased NMDAR in the extrasynaptic space was also observed. However, the density differences in post-synaptic markers are mostly gone by month 6, and they are instead replaced by changes in the area of inhibitory pre-synaptic sites that may be part of a compensatory mechanism. The idea that molecular changes happen at the synapse before we can describe observable symptoms is interesting.

Finally, the authors perform RNA-seq on the SNS of 1 and 6-month old mutant mice to identify differentially expressed RNAs at the synapses. Most of the DEG happen at the 6-month old, with some of them then being direct targets of FUS that were identified by CLIP-seq. The authors conclude that mislocalization of FUS changes the synaptic transcriptome and ultimately synaptic function, through direct and indirect mechanisms. However, notable protein changes were observed at 1 month-old, whilst most transcriptional changes are dominant at 6 months. The authors do not address this issue with enough depth or provide a potential explanation, whilst densities/areas of differentially expressed FUS targets are not explored.

In general, the study addresses relevant points for the function of FUS in ALS. The proof that FUS binds synaptic-protein encoding RNAs is very interesting, as is the identification of those specific RNAs. However, certain methods central to the findings require further validation, whilst the interesting themes of the paper have limited experimental evidence to support the direct connections that have been concluded.

I have the following comments:

Major:

Although it is understood that no protocol is perfect and compromises need to be constantly made, some more clear justification of some aspects would help the reader and make the findings of this study stronger. For example, one of the main points of the study is based on the results from CLIP-seq from SNS. SNS preparation is a harsh and lengthy process that might disturb FUS binding to some of its targets, whilst its limited sequence specificity may allow FUS to engage with RNAs during the processing to which it wouldn't normally under physiological conditions. As the RNAs in the preparation are already enriched for synaptic genes, it may not be surprising that the targets end up being synaptic candidates. A more intuitive approach might therefore be to first perform the cross-linking prior to the SNS preparation. However, there is no justification in the text for having used the current protocol, or support that the CLIP-seq binding profiles aren't an artefact initiated during the tissue processing. Indeed, this offers an alternative explanation to the CDS enrichment the SNS fraction that can't yet be ruled out. This is particularly important to address given CLIP is being used in this study to identify putative FUS targets without any functional validations, and that no replicate datasets are evident. Whilst it is appreciated that SNS from a large number of young mice has been required for each 1-month dataset, either additional replicates should be provided to demonstrate reproducibility, or tests comparing crosslinking pre- and post- SNS purification should be carried out using more abundant adult tissue. For the latter it is recommended that at least one sequence-specific RBP is tested alongside FUS. Should the latter show no evidence of processing induced artefacts, then this would provide sufficient justification for the workflow used and increase the confidence in the present results.

Another example is the reason to apply a filtering for synaptic enriched FUS targets. This filtering excludes FUS targets that might have an important synaptic function despite not being enriched at the synapse. This may include components of the translation machinery or

elements of the cytoskeleton that are more evenly distributed across the cell than synaptic enriched genes. The common targets will be represented in the final list only if the number of peaks is higher (binding of FUS is more frequent). Discussing this limitation would be merited, as would highlighting some potential examples of this that are evident in the lists from less stringent p-value cut-offs.

The connection between manuscript themes is presently limited. For example, putative FUS targets identified by CLIP are up-regulated in SNS fraction at 6 months. However, there is presently no evidence that this manifests in a change in the protein level expression of these genes that then contributes to the synaptic abnormalities reported. Whilst assessment of presynaptic and post-synaptic markers that are putative FUS targets has been carried out (SNAP25, GluN1, GABAAalpha1), these showed no significant changes in figure 3 and were not among the FUS targets changed in the RNA-seq. It is unclear whether these candidates were selected before the RNA-seq results were available, but assessing density/areas of some other putative FUS targets that are differentially expressed should be addressed alongside the current presynaptic and post-synaptic markers in figure 3. If evident, this would provide a stronger mechanistic connection between the themes to support the conclusions made in the manuscript.

Minor:

Fig. 1f and 1g: Does each point represent a discrete FUS signal, synapse, a cell or a stack? Please, specify in the legend. If a synapse, how many cells were used for this analysis?

Fig 2c: qPCR distributions for multiple pre-mRNAs should also be included. This should include Grin1 pre-mRNA. This will be particularly important given the intronic CLIP signal seen in panels 2h and 2i.

Figure 2d: Red boxes are not explained in legend. This is presumed to be the excised region, and if so this information should be added. Reasons for not including low mobility complexes should be addressed.

Figure 2h and 2i: The presence of intronic crosslinking at the synapse, particularly in 2h, should be addressed. It might be helpful to add a colour scale to the peaks which highlights those passing the filter in figure 2e. The current peak track is too small and difficult to distinguish.

Line 254: "On the exon of Grin1" – the specific exon should be listed.

Genera: Are sequences overlapping SNS-enriched FUS CLIP clusters highly conserved?

Line 277-288: From a sequence enrichment analysis in synaptic FUS targets, the authors conclude that FUS does not have a different sequence preference in the synapse than in the nucleus. However, this paragraph on the results section (line 277 to 288) is perhaps a bit confusing/misleading as presently written. The title suggests a new finding of a new binding site exclusively in synapses. However, in the last sentence they state that they do not see a difference in enriched binding sites in the synapses vs. total tissue. Meanwhile, from table 1, there seems to be an enrichment for the sequence AGGUAAR at the SNS, which is different from the one in total tissue, which is not a GU-rich (more purine-rich), and which is present in introns, exons and 5'UTRs. The authors do mention that this motif is only found in a small percentage of the transcripts, and therefore not a strong preference to a sequence. However, it is both as frequent in the targets and as rare in the background as the more GU rich sequences. Whilst there is room for interpretation by the authors, the message from the text needs to be made clear and consistent throughout this section – i.e. either no new FUS binding motif has been found or a new binding motif has been found.

Fig. 3g, h and i. Where possible, it would be easier for the reader if they use the same graph type in all 3 panels.

Fig. 3. The title: “Increased synaptic FUS localization in FUS Δ NLS/+ mice affect GABAergic synapses” is misleading. It implies a causality that is not demonstrated in this figure/results. The authors observe an increase in synaptic FUS localization and in other synaptic markers but they do not demonstrate that the second is a direct consequence of the first. It could be, for example, due to a loss of nuclear FUS instead.

Lines 384:388, methods, Supp figure 4D/E: Details of the methods used for gene ontology analysis should be included. This should provide detail of whether a background set of genes was used in the analysis or not.

Figure 4c: Most of the significant DGE calls have very small effect sizes, and fold-change cut-offs have not been used like for CLIP-seq. Whilst consistent direction of change of course increases confidence, has a power analysis been performed to confirm that calling a $\log_{2}FC > 0$ is possible with the read depths and number of replicates used? Please can the response detail this accordingly. It would also be beneficial to the reader if the range of reads per replicate could be added to the methods when describing the sequencing parameters, or a table of sequencing metrics added.

Line 401: Whilst the overlap suggests potential direct effects of FUS, the CLIP data provides only evidence of a biochemical interaction and not evidence of a functional change to any of the targets identified. This would require mutagenesis / gene editing to confirm that the identified FUS binding sites are functional by correcting changes in gene expression. The text should be adjusted to reflect this accordingly. Note, it is acknowledged that line 483-4 of the discussion already minimally highlights this.

Line 421-422 / 441-422: It would be beneficial to the reader if a brief summary of the behavioural dysfunctions and associated timelines are provided in this manuscript at earlier points. At present, limited information is provided on lines 465-467 and 478-480 after earlier references to the co-submitted manuscript.

Line 477: This sentence needs further elaboration.

Line 560. “REF” ?

Methods cDNA and Quantitative Real-time PCR and Fig. 2c: Methods section says “Total RNA was...” but in the figure we also see a SNS sample. How was the SNS sample prepared? Were those the same samples/brains used for sequencing and/or imaging? In the figure, what does each dot represent? Was any statistical test performed on those qRT-PCR results?

Line 635: The antibody code should be included

Line 640-641: The methods imply 15 cycles of HiSeq 2500 sequencing were performed for the CLIP-seq libraries. As this would lead to reads that are typically too short to map, can the details be checked and updated accordingly.

Fig. 3c legend: please, specify error bars and statistics.

Supplementary Fig. 3b legend: Please, specify error bars and statistics.

Line 1269 (Supp fig 4e legend): “significantly increased RNAs...” but in the figure it says

downregulated.

Supp fig 4f legend: mentions samples at 6-months but does not refer to 1-month.

Reviewer #3 (Remarks to the Author):

There is still a need for a stronger or novel therapeutical approaches for ALS and FTD. Therefore, uncovering novel roles or broadening current understanding of function of disease relevant proteins, such as FUS, is of utmost importance and is of high interest for the field. In this paper the authors first localize FUS to synapses using super resolution microscopy and then using CLIP-Seq on synaptoneurosomal fractions, determine the RNA interactome of synaptic FUS. They show preference of binding to mRNAs associated with synapse organization and plasticity. Using a mouse model of FUS (characterization of which is contained in the co-submitted manuscript Scekcic-Zahirovic et al.) they show age related misregulation of the GABAergic network. The paper is clear, structured and well written, the methodology is appropriate for the conclusions reached.

Of importance, most of the major findings that are discussed in the discussion section point to observations in Scekcic-Zahirovic et al. co-submitted manuscript, which from the behavioral perspective seem to parallel the molecular changes observed in this paper. Therefore, the fate of this manuscript seems to be very much bound to the Scekcic-Zahirovic et al. co-submitted manuscript and the full value of the publication will depend on joint acceptance.

Comments:

- L130-138 The study is focused on FUS related changes in the synapses of the hippocampus, so the authors should present more caution in drawing parallels with the cosubmitted study, whose behavioral outcomes point to brain regions associated with FTD.
- L177. What percentage of FUS colocalizes with spinophilin? How close is this colocalization? Just by eyeball it seems quite significant. Figure 1e should also have an overlay and quantification for spinophilin.
- L182. What is the comparison synapsin 1 with spinophilin?
- Fig1h is slightly at odds with SupFig1d Imaris image, which shows a lot more postsynaptic FUS signal than what is presented in the F1h (only two spots).
- SupFig 1d and h. It would be clearer if the colors in the scheme had a 'mini-legend' next to the scheme. Though one can follow the green FUS, the number of different labelings makes other marker proteins harder to follow.
- SuplFig2a. figure needs correcting. 'Post' is written across one of the blots.
- L273. Superfluous sentence
- L404. Considering their importance for neurodegeneration the authors should comment the changes observed for APP (and APLP1) as is shown in supl fig5.
- L629 Add a reference for the CLIP-seq method.

Reviewer #1

In this manuscript, Saharana et al. molecularly characterize a mutant FUS knock-in mouse model mirroring the heterozygous genotype of human patients with FUS mutations. This work focuses on extranuclear functions of FUS, particularly at the synapse, and reports FUS enrichment at presynaptic sites based on super-resolution microscopy data. Purifying SNS fractions from mouse brain, the authors use CLIP-seq to identify the synaptic RNA targets of FUS and cross reference these with differentially expressed genes between FUS mutant and WT mice forebrain, revealing abnormalities in expression of genes critical for maintaining synapses. Immunostaining experiments show alterations in levels and distribution of many of the gene products from these proteins, supporting dysregulation of these pathways.

While FUS has been implicated in the transport and translation of RNAs in neuronal processes, there is relatively little known about its targets or their impact on neuronal physiology. As such, this manuscript provides new information regarding the potential function and impact of extranuclear FUS function that may have broad relevance for FUS-mediated ALS and FTD. The manuscript is also well written and the figures beautifully illustrated.

We gratefully acknowledge the referee's positive assessment of our work.

Despite this, however, as it stands the manuscript is largely descriptive, and there is limited tie-in between transcriptomic findings and functional consequences. Moreover, methodological and technical caveats complicate the interpretation of data, as outlined below.

We appreciate this concern of the referee and we hope that the additional experimentation and data added in our revised manuscript will be convincing.

Major comments:

- 1. While intriguing, many of the findings are largely descriptive without a clear connection between synaptic FUS, differentially expressed genes/transcripts, and/or downstream functional effects. The demonstration that synaptically-localized FUS binds RNAs related to excitatory and inhibitory synapses is potentially exciting; however, the impact of such binding is unclear, since few differentially expressed genes are direct FUS targets, and few are verified at the protein level.*

We thank the referee for raising this point. We agree that the discovery that synaptically localized FUS binds RNAs related to excitatory and inhibitory synapses is an exciting finding. And indeed establishing clear connections between synaptic FUS RNA binding and specific functions is very important, but also extremely challenging. Unveiling the full picture of the functional consequences of synaptic FUS RNA regulation requires extensive work, beyond the scope of this manuscript.

Nevertheless, the referee's comment encouraged us to perform additional experimentation and bioinformatic analysis and we are now reporting that indeed a clear pattern of FUS-mediated regulation has emerged from this additional work. To address this important point, we performed additional experiments to explore the effect of cytoplasmic FUS on RNA stability and protein levels of its targets. In our new **Supplementary Figure 4e**, we show our proteomic analysis comparing synaptoneurosome extracts from synaptoneurosome of 6-month-old WT versus *Fus*^{ΔNLS/+} mice. This analysis that included six biological replicates of each group confirmed a significant ~3-fold increase of FUS protein in the synaptic sites of *Fus*^{ΔNLS/+} mice, in agreement with our biochemical and image analysis shown in Figure 3a-d. However, while some trends were detected in these experiments, no other significant changes were found at this time point, a result that was confirmed in three independent mass spectrometry experiments. Our interpretation of these data is included in our revised manuscript (lines 342-356). In summary, we think that alterations in protein levels at this early time point, if any, are too mild to be detected by this mass spectrometry approach. Nevertheless, our imaging approach shown in **Figure 3** revealed some significant changes in synaptic organization, which become more prominent with age, as shown in the accompanying paper submitted by Dupuis and colleagues (Fig. 6).

In addition, we have further explored the mechanism by which synaptic FUS accumulation leads to significant changes in synaptic RNA profile. In our new **Figure 5**, we show that FUS is involved in the stabilization of several RNAs, including most importantly its synaptic RNA targets. For this, we followed an unbiased high-throughput sequencing approach to identify the RNAs that show altered stability in neurons from *Fus*^{ΔNLS/+} compared to WT. After treating primary neurons with actinomycin D to block transcription, samples were collected at various time points (0h, 8h, 12h or 24h) for RNA extraction and sequencing, to determine changes in their half-life or decay over time. Importantly, RNAs with increased levels in *Fus*^{ΔNLS/+} at 12 hours were enriched for synaptic localization and function (new **Fig. 5c**), even though the stability assay was performed with total extracts and not synaptoneurosome. In contrast, decreased RNAs at 12h encoded mostly proteins involved in protein translation and RNA binding (new **Supplementary Fig. 9b**). The enrichment of synaptic RNAs among the total more stable RNAs suggested that FUS misaccumulation increases the stability of synaptic RNAs.

We then performed a number of analyses to uncover any potential patterns between FUS targets, localization of the binding and alterations in RNA. Out of 307 synaptically localized FUS RNA targets, 86 show altered stability (new **Figure 5**), while 33 show distinct accumulated levels at the synapse (new **Figure 4**) in *Fus*^{ΔNLS/+} mice. This overlap seems low, but it is important to clarify that we did not anticipate a perfect overlap between the targets and the affected RNAs for a number of reasons. Firstly, we expect that FUS binding is involved in a number of functions, including stability, transport and local translation, not all of which will show altered RNA levels. Secondly, some of the directly affected RNAs and proteins will have secondary effects on more RNAs, generating a number of indirectly regulated RNAs. Lastly, while our aim was to focus on the early changes in *Fus*^{ΔNLS/+} mice, it is important to emphasise that some of the direct effects of FUS binding may only be apparent later in disease.

The overlap is minimal and without statistical enrichment between FUS targets and decreased RNAs, both at 6-month-old mice and at our stability assay. However, our new analysis showed that there is a significant enrichment of FUS targets among the accumulated RNAs at 6 months, which remains significant if we include the group of RNAs that are collectively increased - but not

decreased - in the stability experiment. We also uncovered a significant enrichment of exonic binding on RNAs that are either increased in stability or accumulated in *Fus*^{ANLS/+} mice and reversely, an enrichment of accumulated or stabilized RNAs synaptic FUS targets with exonic binding. We propose that a subset of synaptic RNA targets is stabilized by FUS binding on multiple exonic sites, while the 3' UTR binding has a distinct function, potentially in regulating local translation. The precise mechanism of how FUS regulates these targets will be the focus of future investigations.

2. Further, it remains unclear whether extranuclear FUS, or a relative reduction in nuclear FUS, is responsible for these differences.

While the referee raises an important point, previous work on this mouse model has convincingly shown that the described phenotype in these mice is not due to a relative reduction in nuclear FUS, but directly linked to the cytoplasmic mislocalization of FUS. Indeed, it was shown previously that there is no reduction in nuclear FUS in the spinal cord of *Fus*^{ANLS/+} mice, using semi-quantitative western blots with two different antibodies raised against either the N-terminal or the C-terminal region of FUS (see Fig 1a-c of Scekcic-Zahirovic et al, 2017: <https://www.ncbi.nlm.nih.gov/pmc/articles/PMC5427169/>). The same conclusion was reached using immunofluorescence analysis (see Fig 1d-e of Scekcic-Zahirovic et al, 2017), which is why the authors concluded that the observed phenotype in these mice is due to cytoplasmic mislocalization of FUS.

3. Similarly, there is no effort to tie together the changes in differential RNA expression observed at 1 or 6 mo with the observed changes in excitatory or inhibitory synapses detected in these animals. These phenotypes are observed and described, without investigating direct or indirect connections.

As described in detail in point 1 above, the focus of our revision was to link the observations of the different parts of our work. In summary, our main conclusions from our new experimentation and bioinformatic analysis are the following:

1. Protein levels of specific RNA targets are likely changed at the level of specific synapse subtypes and are therefore, not detectable at a proteomic analysis of total cortical synapses.
2. Synaptic RNAs that are accumulated at 6 months of age are significantly enriched for direct FUS targets, while those that are decreased aren't.
3. There is a significant correlation between the RNAs that are accumulated in 6-month-old mice and those that are increased in short term RNA stability.
4. Increased RNAs (either accumulated at 6 months, or increased in stability) are significantly enriched for exonic binding of FUS, frequently with multiple exonic binding sites.
5. Reversely, there is an enrichment of synaptic RNAs with increased accumulation at 6 months among FUS targets with exonic binding.
6. We conclude that a subset of synaptic RNA targets is stabilized by FUS binding on multiple exonic sites.

7. 3' UTR binding shows a novel binding motif, distinct from the exonic motif and is potentially dependent on secondary structures on target RNAs recognized via the RRM of FUS, in line with recent structural studies.
 8. We hypothesise that 3' UTR binding has a distinct function, potentially in regulating local translation. The precise mechanism of how FUS regulates these targets will be the focus of future investigations.
4. *What is the proportion of FUS within synapses vs. total FUS in the cell? Images in Fig. 1 suggest that the pool of synaptic FUS is minimal, with the signal near background levels. With signals this low, negative controls are a must—these might include FUS KO tissue, for instance—to increase confidence in the staining.*

We agree with the referee that synaptic FUS is only a small fraction of the total cellular FUS. In fact, we were able to quantify this difference, using our biochemical analysis from synaptoneurosomes versus total extracts and we estimate synaptic FUS to be roughly 10% of total FUS. This new quantification shown below is now included in our new **Supplementary Figure 2b**). Despite this uneven proportion and the low signal recorded in the synapse by our imaging analysis, we can exclude that the signal is background for the following reasons. First, the FUS antibodies that we used (Rb, A300-293A, Bethyl and Ms, 4H11, Santa Cruz) was previously validated by the Dupuis team and has shown no background signal in FUS KO tissues - see Scekcic-Zahirovic et al, 2016 study, Figure 3. Moreover, in our **Figure 1**, we used Ground State Depletion (GSD) imaging coupled with dSTORM to analyze the distribution of FUS within the synapse. GSD imaging requires a bleaching step of the sample to obtain the blinking state of the fluorescent probes. This ensures absence of unspecific background and acquisition of specific signals. Moreover, in the post-processing using ThunderSTORM, collected particles were filtered for the final image reconstruction in order to keep only the highly precisely localized particles.

5. *This manuscript finds very few changes in RNA abundance in total forebrain, while the cosubmitted manuscript using the same mouse model finds hundreds of transcriptomic changes in the frontal cortex. What accounts for this large discrepancy?*

We thank the referee for raising this point, which encouraged us to reanalyse the sequencing data from both studies side by side and by using identical analysis pipelines. The result was very clear. The discrepancy can be explained by differences in the analysis pipelines, but not on biological differences, which further validated the independent datasets from both groups. To understand the origin of the apparent discrepancy, we exchanged datasets and analysis pipelines between both labs. The Dupuis team previously used a DEseq-based model, while we used a more recent and more conservative EdgeR-based model. The difference between both pipelines is illustrated in the QQ plots below, that show the correlation between expected and observed p-values for the same dataset using either the Scekcic-Zahirovic et al pipeline (left) or the Sahadevan et al pipeline (right). The ideal calibration is expected to lie within the grey zone. The Scekcic-Zahirovic et al pipeline (left) tended to inflate the observed p-values as compared to expected p-values, which resulted in an overestimation of the number of differentially expressed genes as compared to the

Sahadevan pipeline, which appeared to be better calibrated on p-values although slightly underpowered. As a consequence, the Scekcic-Zahirovic et al bioinformatics analysis likely included some false positives. In contrast, the Sahadevan et al pipeline was more stringent, avoiding false positives, but potentially missing some differentially expressed genes that would not be identified as significant. In order to avoid the discrepancy between the two papers, we decided to use the exact same conservative pipeline of analysis for the identification of differentially expressed genes between both manuscripts, i.e. the same one that we used in our originally submitted manuscript.

While our results and conclusions on this point remain unaltered in our revised manuscript, the revised Scekcic-Zahirovic et al study has now replicated our findings using their own dataset, and did not identify differentially expressed genes. They then used a systems biology approach to define altered gene network nodes. The updated conclusion of this analysis is detailed in the Scekcic-Zahirovic et al response to the referee and manuscript. Importantly and in agreement with our study, this new approach suggests a widespread defect of synapses in the *Fus* Δ NLS/+ mice. Summarizing, we would like to thank the reviewer for pointing out this discrepancy between our jointly submitted manuscripts. The harmonization of the analytical bioinformatic pipelines validated the results between both laboratories, and independently confirmed the key role of synaptic misregulation in both studies.

6. While intronic reads are certainly de-enriched, it appears that nearly 20% of the FUS binding sites identified by CLIP in the SNS preparations are in introns, bringing up the possibility of nuclear contamination in the SNS preparations.

We thank the referee for bringing up this important point. We have now looked deeper within the synaptic FUS targets with apparently intronic localization and we present our analysis in our revised manuscript. We have considered and explored the possibility that all these targets

originate from nuclear contamination, since no biochemical isolation, no matter how pure, is ever totally devoid of traces of contaminating material. This point was clearly stated in our originally submitted manuscript, even though we have thoroughly assessed the purity of our synaptoneurosomes in **Figure 2** and found no evidence of contamination.

In our new **Supplementary Figure 3**, we utilized Ribo-Zero RNA-seq data from total cortex and from our synaptoneurosomes in order to better assess the level of possible nuclear spillover in our samples. Comparing the levels of the exclusively nuclear non-coding RNA *Malat1* showed no signs of contamination in our synaptoneurosomes sample, as did comparing the intronic coverage of one of our synaptic targets *Grin1* (**Supplementary Figure 3b-c**). We then closely inspected some of our synapse-specific intronic peaks and wondered if they might be consistently low coverage peaks, maybe falsely called by our analysis pipeline. This was excluded by our comparison of the maximal peak length between our intronic peaks and all other synapse-specific peaks, which showed no difference (**Supplementary Figure 3a**), further validating our analysis pipeline. We then looked through several of these peaks manually to identify any potential commonalities or irregularities and we report at least two “classes” of such peaks. One, exemplified by *Slc9a9* (**Supplementary Figure 3d**) is a strong nuclear target of FUS with multiple nuclear intronic binding. In this case, we cannot exclude a potential spillover from nuclear FUS complexes, although we note that the synapse-specific peak is very distinct from the total sample and might represent unannotated elements, such as exons or non-coding RNAs. In fact, that seems most likely to be the case in our example of *Peak1*, which shows a very specific binding, annotated as intronic, but actually overlapping with the miRNA *Gm24270* (**Supplementary Figure 3e**). Indeed, we show that the region corresponding to the pre-miRNA of *Gm24270*, but not the rest of *Peak1* intron, is consistently expressed in all our sequencing datasets, including Ribo-Zero and even poly(A) RNA-seq of synaptoneurosomes. This suggests that, at least in some cases, the apparently intronic FUS binding, actually represents binding on overlapping non-coding RNAs and is not due to nuclear contamination.

7. *Related to this, with such a small proportion of total FUS in the SNS, how are the read counts in the SNS preps (Fig. 2) just as high as in the total cellular preps?*

We thank the referee for the opportunity to clarify this point. The reason for this apparent discrepancy is that the two libraries (total or SNS CLIP-seq) were sequenced in approximately the same depth, yet we strongly enriched for synaptic RNAs in our SNS preparation. Therefore, a direct comparison between the CLIP-seq samples is not valid, as the synaptic targets will be diluted or even missed in the total CLIP-seq library, while they will be overrepresented in the SNS CLIP-seq. This was exactly the reason why performing the SNS CLIP-seq was necessary in the first place and it wouldn't have been possible to predict the synaptic RNA targets of FUS from previously available total brain CLIP-seq data.

8. *Given differences between mouse and human, the authors are encouraged to examine FUS distribution in human tissues as well as mouse, thereby confirming conservation of FUS distribution in humans*

We thank the referee for raising this point and we agree that comparing the subcellular distribution and behavior of FUS protein in mouse and human neurons is extremely important for extrapolating conclusions from this mouse model to human disease. Several previous studies have addressed this point and found no differences between mouse and human FUS in neurons. The distribution of FUS in the human brain has been extensively studied since ALS mutations in FUS were discovered (Kwiatkowski. et al.; Vance et al., Science 2009) and no significant difference between species have been reported. It is also important to note that the amino acid sequence of human and mouse FUS proteins are 95% identical and the nuclear localization signal is 100% identical between the two species. Moreover, in a previous study, our team has systematically compared the subcellular distribution of FUS in mouse and iPSC-derived human neurons, under control conditions or cellular stress and we found no differences between species (Hock et al Cell Rep, 2018). In full agreement with our current study, synaptic localization of FUS in human neurons and synaptic accumulation of mutant FUS in iPSC-derived neurons from ALS patients was also confirmed in a recent study (Deshpande et al., 2019).

Minor comments:

Line 91: should be “in sporadic FTD”

We thank the referee for pointing out this error. This sentence has been updated in the revised manuscript in line 90 as *“However, in sporadic FTD, FUS mislocalization occurs in the absence of mutations.”*

- *Line 161: references are needed for using VGAT and PSD95 as markers of inhibitory and excitatory synapses, respectively*

We thank the referee for pointing this out. The references are now added in the line 166 of the revised manuscript. *“VGAT was used as a marker for inhibitory synapses⁴⁵ and PSD95 for excitatory synapses⁴⁶”*

- *Line 164: the authors’ conclusion that “extranuclear FUS preferentially associates with excitatory synapses” is circular, since they only included FUS clusters within 200 nm from a synaptic marker, and therefore cannot comment on the entire pool of extranuclear FUS*

We thank the referee for pointing this out and we agree that the word “extranuclear” was inaccurate in this case. In our revised manuscript, we have exchanged the word “extranuclear” to “synaptic”.

- *Line 188: these results are contradictory, but do show that FUS is present in both pre- and post-synaptic regions, although more so in the former.*

We agree with the referee that our data indicate the presence of FUS at both pre and postsynaptic sites and we clearly state this in the following statement: *“These results are in line with the previously published localization of FUS within 150 nm from the active presynaptic zone²⁷, but highlight the presence of FUS also at the postsynaptic site, potentially explaining the apparently contradictory results of previous studies^{20,28}.”*

- *Line 299: given the nature of the FUS Δ NLS knock in model and the loss of one allele encoding full-length FUS, wouldn't the authors expect an overall reduction in full-length FUS, regardless of localization?*

Yes, we definitely agree with this point. Indeed, we see a reduction of full length in the *FUS Δ NLS* knockin model. This is shown in **Figure 3 b-c** and **Figure 4 a-b**.

- *Line 353: "instable" should be "unstable"*

We thank the referee for pointing this typo, which is now corrected in our revised manuscript.

- *Is the distribution of FUS any different within spinal motor neurons, vs. cortical or hippocampal neurons?*

This is an interesting question that we have considered in depth. The Dupuis team has previously reported mislocalization of FUS in several cell types, including motor neurons (Scekic-Zahirovic et al, 2016, 2017), oligodendrocytes (Scekic-Zahirovic et al, 2017) and muscle cells (Pichiarelli et al, 2019). In addition, our analysis in the current study included both hippocampal and cortical neurons and showed the expected FUS mislocalization in the *FUS Δ NLS* mice without any observed differences between the cell types. Taken together and considering the ubiquitous expression of FUS and the conservation of the import mechanism that is disrupted by the deletion of the NLS in these mice, we consider it rather unlikely that there are cell-specific differences in FUS distribution.

Figure comments-

Fig 1:

- *a- please mention the use of PNF as neuronal marker in the text*

We used PNF as a neuronal marker to stain the axons. Indeed, we have mentioned in the lines, 159-160. *“.... which confirmed the presence of extranuclear FUS clusters along dendrites and axons (identified with MAP2 and PNF, respectively)”*

- *h- please label pre- and post-synaptic regions, as well as yellow dots*

Thanks for pointing out these errors. Yellow dots have been removed as this was an error. Figure 1h has been updated with pre- and post-synaptic regions labelled in the schematic.

Fig. 2:

- *The MA plots in Fig. 2e-f each displays a row of points in the upper left corner distinct from the rest of the data, but the significance of these rows is unclear.*

We thank the referee for pointing this out. The row of points in the upper left corner are the peaks with zero reads in the total cortex sample. They separate from the rest of the points because of the prior count (0.125), which is required to prevent a division by 0 when computing the log₂ fold change for these peaks. We have updated the figure legend accordingly and included the following sentence: *“The row of points in the upper left corner are peaks with zero reads in the total cortex sample.”*

Sup Fig. 4:

- *f, g- while the heat map displays many potential DEGs, the volcano plots do not show this.*

Indeed, the heatmap for total cortex (**Supplementary Fig. 5f**) shows similar trends as the heatmap for SNS: decreased RNAs show a trend for negative log fold changes and increased RNAs show a trend for positive log fold changes. However, the trend is not as consistent across all replicates and there is more variability than in SNS. In addition, the log fold changes in both plots are not the same. The heatmap shows the log fold changes of each individual sample compared to the mean expression of the *FUS*^{+/+} samples at each time point, while the log fold changes in the volcano plot are based on the mean expression of all replicates.

Reviewer #2

In this study by Sahadevan et al., the authors aim to gain further knowledge about FUS protein functions at the synapse regulating RNA homeostasis and the implications of ALS associated mutations at the synaptic level.

First, they confirmed previously described synaptic localization of FUS, in both excitatory and inhibitory synapses, with a preference at the presynaptic terminal but also present at the postsynaptic site. Second, they aimed to identify FUS-bound RNAs at the synapses. In order to enrich for synaptic localized RNAs the authors prepared synaptoneuroosomes (SNS) upon which Crosslinking Immunoprecipitation (CLIP) was performed. The integrity of the SNS fraction is shown by imaging, western blot enrichment for synaptic proteins, and qRT-PCR for synaptic mRNAs. After SNS-CLIP-seq using a single replicate derived from pooled samples, a filtering is applied to identify synaptic exclusive FUS RNA targets. In the 307 RNAs selected, FUS mostly binds to exons and 3'UTRs in contrast to the preferent binding to introns in the whole tissue sample/whole cell. Within FUS synaptic targets there is an enrichment for those involved in transport, localization, trans-synaptic activity, signalling, receptor binding and transmembrane

transporter activity. However, this result might not be surprising since those are terms associated to synaptic activity in a list enriched for synaptic protein encoding RNAs.

To study the implications of ALS mutant FUS in synaptic function, the authors use mice lacking the nuclear localization signal of FUS, which accumulates FUS at the cytoplasm, similar to what happens with ALS-associated FUS mutations. The truncated protein not only accumulates at the cytoplasm, but it is also enriched at the synapses to suggest a potential misregulation of the aforementioned synaptic targets could occur. To explore the consequences of FUS mislocalization, the authors performed IHC and imaging for markers of different synaptic structures, at 1 and 6 months old. They observe an increase in density of NMDAR and a decrease in density of GABAAalpha3 in post-synaptic sites at 1 month suggesting a synaptic hyperexcitability. Increased NMDAR in the extrasynaptic space was also observed. However, the density differences in post-synaptic markers are mostly gone by month 6, and they are instead replaced by changes in the area of inhibitory pre-synaptic sites that may be part of a compensatory mechanism. The idea that molecular changes happen at the synapse before we can describe observable symptoms is interesting.

Finally, the authors perform RNA-seq on the SNS of 1 and 6-month old mutant mice to identify differentially expressed RNAs at the synapses. Most of the DEG happen at the 6-month old, with some of them then being direct targets of FUS that were identified by CLIP-seq. The authors conclude that mislocalization of FUS changes the synaptic transcriptome and ultimately synaptic function, through direct and indirect mechanisms. However, notable protein changes were observed at 1 month-old, whilst most transcriptional changes are dominant at 6 months. The authors do not address this issue with enough depth or provide a potential explanation, whilst densities/areas of differentially expressed FUS targets are not explored.

In general, the study addresses relevant points for the function of FUS in ALS. The proof that FUS binds synaptic-protein encoding RNAs is very interesting, as is the identification of those specific RNAs. However, certain methods central to the findings require further validation, whilst the interesting themes of the paper have limited experimental evidence to support the direct connections that have been concluded.

We thank the reviewer for the accurate summary of our study, his/her overall positive assessment and the constructive comments, which helped us revise and improve our work.

I have the following comments:

Major comments:

- 1. Although it is understood that no protocol is perfect and compromises need to be constantly made, some more clear justification of some aspects would help the reader and make the findings of this study stronger. For example, one of the main points of the study is based on the results from CLIP-seq from SNS. SNS preparation is a harsh and lengthy process that might disturb FUS binding to some of its targets, whilst its limited sequence specificity may allow FUS to engage with RNAs during the processing to which it wouldn't normally under physiological conditions. As the RNAs in the preparation are already*

enriched for synaptic genes, it may not be surprising that the targets end up being synaptic candidates. A more intuitive approach might therefore be to first perform the cross-linking prior to the SNS preparation. However, there is no justification in the text for having used the current protocol, or support that the CLIP-seq binding profiles aren't an artefact initiated during the tissue processing. Indeed, this offers an alternative explanation to the CDS enrichment of the SNS fraction that can't yet be ruled out. This is particularly important to address given CLIP is being used in this study to identify putative FUS targets without any functional validations, and that no replicate datasets are evident. Whilst it is appreciated that SNS from a large number of young mice has been required for each 1-month dataset, either additional replicates should be provided to demonstrate reproducibility, or tests comparing crosslinking pre- and post- SNS purification should be carried out using more abundant adult tissue. For the latter it is recommended that at least one sequence-specific RBP is tested alongside FUS. Should the latter show no evidence of processing induced artefacts, then this would provide sufficient justification for the workflow used and increase the confidence in the present results.

We thank the referee for this thoughtful and constructive criticism, which we addressed with additional experimentation, exactly as suggested. We agree with the referee that since FUS is such a promiscuous binder with very little sequence specificity, it is hard to conclusively validate our approach for the synaptic RNA binding solely by the SNS FUS CLIP-seq. To address this point, we performed CLIP-seq for TDP-43 in synaptoneurosomes, following the exact same approach that we followed for FUS. We consider TDP-43 to be the perfect control here because, like FUS, its synaptic levels are considerably lower than its nuclear levels, but, unlike FUS, TDP-43 has a well characterized sequence specificity to GU repeats (Polymenidou et al, Nat. Neurosc 2011; Lukavsky et al, NSMB, 2013). As shown in our revised **Table 1**, in our SNS TDP-43 CLIP-seq we find a strong enrichment for UG repeats, following an identical protocol that we used for FUS. We are therefore confident that our SNS CLIP-seq protocol yields specific RBP-RNA binding profiles and that the motifs identified in SNS FUS CLIP-seq are valid.

- 2. Another example is the reason to apply a filtering for synaptic enriched FUS targets. This filtering excludes FUS targets that might have an important synaptic function despite not being enriched at the synapse. This may include components of the translation machinery or elements of the cytoskeleton that are more evenly distributed across the cell than synaptic enriched genes. The common targets will be represented in the final list only if the number of peaks is higher (binding of FUS is more frequent). Discussing this limitation would be merited, as would highlighting some potential examples of this that are evident in the lists from less stringent p-value cut-offs.*

We agree with the referee that our analysis excludes FUS targets important for synaptic function that are not enriched at that synapse. While such targets are certainly important, our previous work has highlighted intronic binding of nuclear FUS within RNAs encoding proteins that are important for synaptic function (Polymenidou et al. Nat. Neurosc. 2011; Lagier-Tourenne et al. Nat. Neurosc. 2012). The goal of the current study was to identify unique synaptic FUS targets

that are important for synaptic function and that are directly bound and regulated by FUS at the synapse. We designed our pipeline knowing that we might miss FUS targets that are bound in both the nucleus and the synaptic site. However, we intentionally focused on the synapse-specific targets because they have not been studied before (in contrast to nuclear targets) and also because we hypothesize that they may be linked to synaptic alterations due to synaptic FUS accumulation early in disease. We added the following sentence in line 229 of our revised manuscript to make it more clear to the reader: “We designed a pipeline to identify FUS binding sites that are specific for the synapse and not bound in the nucleus.”

3. *The connection between manuscript themes is presently limited. For example, putative FUS targets identified by CLIP are up-regulated in SNS fraction at 6 months. However, there is presently no evidence that this manifests in a change in the protein level expression of these genes that then contributes to the synaptic abnormalities reported. Whilst assessment of presynaptic and post-synaptic markers that are putative FUS targets has been carried out (SNAP25, GluN1, GABAalpha1), these showed no significant changes in figure 3 and were not among the FUS targets changed in the RNA-seq. It is unclear whether these candidates were selected before the RNA-seq results were available, but assessing density/areas of some other putative FUS targets that are differentially expressed should be addressed alongside the current presynaptic and post-synaptic markers in figure 3. If evident, this would provide a stronger mechanistic connection between the themes to support the conclusions made in the manuscript*

We thank the referee for this valid critique, which was also raised by the other reviewers. Indeed, better connecting the themes of the different parts of our work was the focus of our revisions. We detail our new analysis above, in our response to points #1 and #3 of the first referee. In summary, we now provide proteomics data from synaptoneurosomes at 6 months of age, which consistently show an increase in synaptic FUS, in agreement with our biochemical and image analysis. However, no other protein alterations between *Fus*^{+/+} and *Fus* ^{Δ NLS/+} were significant in this analysis. We discuss the possible reasons for the lack of alterations in our revised manuscript (lines 342-356) and our response to point #1 of the first referee above. Most likely, the protein levels of specific RNA targets are changed at the level of specific synapse subtypes and are therefore, not detectable at a proteomic analysis of total cortical synapses. Indeed, in our image analysis (Figure 3; Supplementary Figure 4 and Supplementary Tables 2-3), which included several FUS RNA targets (SNAP25, GluN1, pCaMKII, GABAalpha1), we saw a slight, but significant increase of GluN1, whose mRNA (*Grin1*) is significantly accumulated at synaptoneurosomes at 6 months. Several other targets showed trends towards increased protein levels at our 6-month image analysis, namely SNAP25, GluN1, pCaMKII and GABAalpha1 (**Supplementary Figure 4g-h**), although these changes did not reach statistical significance. Since we did not specify the type of synapse in our image analysis, it is possible that some of these changes are specific to one synapse subtype and are therefore missed in these quantifications.

While these data suggest mild, but potentially biologically meaningful alterations in specific synapses, we then focused on understanding and dissecting the direct effect of synaptic FUS

binding in altering the RNA profile at the synaptic sites. The main conclusions of this analysis are listed below:

1. Synaptic RNAs that are accumulated at 6 months of age are significantly enriched for direct FUS targets, while those that are decreased aren't.
2. There is a significant correlation between the RNAs that are accumulated in 6-month-old mice and those that are increased in short term RNA stability.
3. Increased RNAs (either accumulated at 6 months, or increased in stability) are significantly enriched for exonic binding of FUS, frequently with multiple exonic binding sites.
4. Reversely, there is an enrichment of synaptic RNAs with increased accumulation at 6 months among FUS targets with exonic binding.

We propose that a subset of synaptic RNA targets is stabilized by FUS binding on multiple exonic sites, while the 3' UTR binding has a distinct function, potentially in regulating local translation. The precise mechanism of how FUS regulates these targets will be the focus of future investigations.

Minor comments:

- *Fig. 1f and 1g: Does each point represent a discrete FUS signal, synapse, a cell or a stack? Please, specify in the legend. If a synapse, how many cells were used for this analysis?*

On the graph, each point represents a field of view used for the quantification. This information is now also added in the respective legend. With the magnification of the STORM, and the imbrication of the cells, it is not possible to determine which synapse relates to which cell. We analyzed fields of view with comparable synaptic density.

- *Fig 2c: qPCR distributions for multiple pre-mRNAs should also be included. This should include *Grin1* pre-mRNA. This will be particularly important given the intronic CLIP signal seen in panels 2h and 2i.*

We agree with the referee that excluding the presence of pre-mRNA or in general nuclear RNA is very important for our conclusions. To address this point, we looked at the intronic coverage of several transcripts in our Ribo-Zero RNA-seq data from synaptoneurosomes and total cortex samples. As shown in our new **Supplementary Figure 3a**, we see very little coverage of *Malat1*, a noncoding RNA known to be enriched in the nucleus. In addition, synaptoneurosome Ribo-Zero RNA-seq shows no intronic coverage in *Grin1* RNA, in contrast to total extracts, showing intronic reads, due to the presence of pre-mRNA present in these samples. This analysis suggested that our SNS preparation is strongly enriched for mature mRNAs that do not have detectable intronic coverage because nuclear RNAs are effectively depleted during our SNS preparation.

- *Figure 2d: Red boxes are not explained in legend. This is presumed to be the excised region, and if so this information should be added. Reasons for not including low mobility complexes should be addressed.*

Thank you for noticing that we missed the description included in the legend. We have now added the following information in lines 1371-1372 of our revised manuscript. *“The red box indicates the excised part used for preparing CLIP libraries”*. The reason for not including low mobility complexes is because they may correspond to heterogeneous or more than one FUS protein complex and hence were not included. This was mentioned in our originally submitted manuscript and is in lines 216-219 of our revised manuscript. *“The autoradiograph showed an RNA smear at the expected molecular weight of a single FUS molecule (70 kDa) and lower mobility complexes (above 115 kDa) that may correspond to RNAs bound by more than one FUS molecule or a heterogeneous protein complex”*

- *Figure 2h and 2i: The presence of intronic crosslinking at the synapse, particularly in 2h, should be addressed. It might be helpful to add a colour scale to the peaks which highlights those passing the filter in figure 2e. The current peak track is too small and difficult to distinguish.*

We apologize that the current peak track is hard to distinguish. This is because we wanted to show the peaks distributed throughout the whole gene. In fact, the peak that is marked as red in the peak track and also marked with red arrows are the peaks that passed the filtering. We have now additionally drawn a red box around the peak to make this even more clear.

We have also extensively addressed the question on the synaptic FUS targets with apparently intronic localization and we present our analysis in our new **Supplementary Figure 3** of our revised manuscript. For a detailed description on this, we refer to our response on point #6 to the first referee above. In summary, we found no evidence of nuclear contamination, although we cannot formally exclude this possibility, as we stated in our manuscript. However, overall these apparently intronic peaks in our synaptoneurosome preparation are correctly called by our pipeline and they may, at least in some cases, represent overlapping non-coding RNAs and not pre-mRNA, as is exemplified for the miRNA *Gm24270* in **Supplementary Figure 3e**.

- *Line 254: “On the exon of *Grin1*” – the specific exon should be listed.*

Thank you for noticing that we missed this description. We have now added the following information in lines 279-281 of our revised manuscript. *“Among those, FUS peaks on exon 18 of *Grin1* (Glutamate ionotropic NMDA type subunit 1) and the 3’UTR of a long isoform of *Gabra1* (Gamma aminobutyric acid receptor subunit alpha-1)”*.

- *General: Are sequences overlapping SNS-enriched FUS CLIP clusters highly conserved?*

The referee raises a question that is particularly relevant for human disease. Since our CLIP-seq protocol does not allow for single nucleotide resolution, we were reluctant to check the conservation on the binding sites that we defined in mouse synaptoneurosome. However, the synaptic FUS targets that we defined are strongly conserved between mouse and human and most of them show >90% at the gene level. Notable examples of this high level of conservation is *GABRA1*, *GABRB3* and *GABBR1*: all ~92% similarity, *GRIA2* 94% and *GRIA3* 95% similarity. We expect that the similarity for the exonic and 3' UTR binding sites will be at least as similar, if not higher, since the coding regions are typically more conserved among species.

- *Line 277-288: From a sequence enrichment analysis in synaptic FUS targets, the authors conclude that FUS does not have a different sequence preference in the synapse than in the nucleus. However, this paragraph on the results section (line 277 to 288) is perhaps a bit confusing/misleading as presently written. The title suggests a new finding of a new binding site exclusively in synapses. However, in the last sentence they state that they do not see a difference in enriched binding sites in the synapses vs. total tissue. Meanwhile, from table 1, there seems to be an enrichment for the sequence AGGUAAR at the SNS, which is different from the one in total tissue, which is not a GU-rich (more purine-rich), and which is present in introns, exons and 5'UTRs. The authors do mention that this motif is only found in a small percentage of the transcripts, and therefore not a strong preference to a sequence. However, it is both as frequent in the targets and as rare in the background as the more GU rich sequences. Whilst there is room for interpretation by the authors, the message from the text needs to be made clear and consistent throughout this section – i.e. either no new FUS binding motif has been found or a new binding motif has been found.*

We thank the referee for raising this important point. Indeed, our data showed a slight deviation from the previously reported motif for FUS in all synaptic binding regions except the 3' UTR group, which showed an even larger deviation, in fact, an entirely new motif: AC-repeats. Knowing that FUS is a promiscuous binder and that these motifs were only found in a small percentage of the targets, we were initially reluctant to make definitive claims about these differences. However, encouraged by this referee's comment and also by our additional analysis on TDP-43, which very consistently identified the well-established UG-repeat, we have now re-evaluated the significance of these different motifs. Structural analyses from our collaborators have shown that the binding motif interacting with the Zinc Finger domain of FUS is actually GGU (Loughlin et al, Mol. Cell 2019). While we cannot formally exclude the significance of the flanking sequences for this binding, we think that it is unlikely to be of great importance. Therefore, we consider the GGU-containing motifs identified in our synaptic target groups to be in line with previous work depicting overlapping sequences. However, the AC-repeat, which was significantly and specifically enriched within the 3' UTR group of synaptic targets suggests a different binding mode, independent from the Zinc Finger domain of FUS. Instead, this sequence is likely suggesting an interaction via the RRM of FUS, which showed affinity for AC sequences within stem loop secondary structures (Loughlin et al, Mol. Cell 2019), as also emphasized by the very recent publication (Jutzi et al., Nat. Comm, 2020). This is now highlighted in our revised manuscript.

- *Fig. 3g, h and i. Where possible, it would be easier for the reader if they use the same graph type in all 3 panels.*

We agree with the referee that it is easier to have the same graph type whenever possible. However, here we chose different representations because the data displayed are different. Bar graphs represent the normalized density of synaptic clusters, which was averaged per brain slice imaged. For the cluster size, we used box and whiskers graph to show the distribution of the size of the clusters. This graph summarizes all the clusters measured across the different mice. Averaging per mouse does not make sense here and bar graph representation would lack information.

- *Fig. 3. The title: “Increased synaptic FUS localization in FUS Δ NLS/+ mice affect GABAergic synapses”. is misleading. It implies a causality that is not demonstrated in this figure/results. The authors observe an increase in synaptic FUS localization and in other synaptic markers but they do not demonstrate that the second is a direct consequence of the first. It could be, for example, due to a loss of nuclear FUS instead.*

We apologize if the title of the figure was misleading. We did not draw the conclusion that the mislocalization of FUS directly affects GABAergic synapses, but we found a correlation between FUS mislocalization and changes in GABAergic synapses. More precise investigation would be required to identify the cellular mechanisms in place. Whether local regulation of RNA by FUS or a nuclear dysfunction are responsible for these changes is not yet known, but both mechanisms are plausible and will be the focus of follow up studies in the lab. However, the levels of FUS in the nucleus remain the same and splicing is not affected, so this mechanism is unlikely to contribute. To avoid confusion, we have rephrased this as follows: *“Increased synaptic FUS localization in FUS Δ NLS/+ mice is accompanied by alterations in GABAergic synapses”*.

- *Lines 384:388, methods, Supp figure 4D/E: Details of the methods used for gene ontology analysis should be included. This should provide detail of whether a background set of genes was used in the analysis or not.*

We apologize for not specifying the details of the GO analysis. The manuscript now includes a sentence in the main text in lines 432-434: *“We performed an over representation analysis on the sets of increased and decreased RNAs and used the genes that are expressed in all FUS^{+/+} SNS samples at 6 months as background. “*

The legend for Supplementary Figure 5d and e was also updated (lines 1556-1558): *“Selected GO terms enriched among the significantly increased/decreased genes at 6 months of age in synaptoneuroosomes of FUS ^{Δ NLS/+} compared to FUS^{+/+} (over representation analysis with all expressed genes in FUS^{+/+} SNS at 6 months as background) “*

We added the following paragraph to the methods section in lines 911-913: *“For the over representation analysis, we applied the “goana” function from the limma R package using the gene length as covariate⁹⁵. As a background set, we used all genes with a cpm of at least 1 in all $Fus^{+/+}$ SNS replicates at 6 months.”*

- *Figure 4c: Most of the significant DGE calls have very small effect sizes, and fold-change cut-offs have not been used like for CLIP-seq. Whilst consistent direction of change of course increases confidence, has a power analysis been performed to confirm that calling a $\logFC > 0$ is possible with the read depths and number of replicates used? Please can the response detail this accordingly. It would also be beneficial to the reader if the range of reads per replicate could be added to the methods when describing the sequencing parameters, or a table of sequencing metrics added*

The reviewer raises an important consideration. We did not perform a power analysis, because we did not know the number and size of the expected gene expression changes between $Fus^{\Delta NLS/+}$ and $Fus^{+/+}$ SNS and the variability between replicates. However, the fact that we did find differentially expressed genes at 6 months and the significant overlap with synaptic FUS targets ensures us that the observed changes are true. We have updated the methods section and included a statement on the sample sizes on lines 899-900: *“After preprocessing, the number of reads per sample ranged from 25 million to 50 million with a median of 35 million.”*

- *Line 401: Whilst the overlap suggests potential direct effects of FUS, the CLIP data provides only evidence of a biochemical interaction and not evidence of a functional change to any of the targets identified. This would require mutagenesis / gene editing to confirm that the identified FUS binding sites are functional by correcting changes in gene expression. The text should be adjusted to reflect this accordingly. Note, it is acknowledged that line 483-4 of the discussion already minimally highlights this.*

We agree with the reviewer that the CLIP data provides only evidence of a biochemical interaction but not evidence of a functional consequence. While the significant overlap of FUS targets among the accumulated RNAs in $Fus^{\Delta NLS/+}$ mice suggests direct and indirect effects of mutant FUS at the synapses, we definitely think that further functional studies are required to fully characterize the implication of the direct FUS binding on its targets. We have updated this sentence in line 452-454). *“While this significant enrichment suggests that FUS binding may mediate the synaptic accumulation of these RNAs, further studies are required to fully characterize the functional implication of the direct FUS binding on its targets”*. Moreover, we have clearly stated in our discussion that further studies are necessary, for example in these sentences of our revised discussion: *“While the precise mechanism of how FUS regulates these targets will be the focus of future investigations, ...”* and *“Dissecting the exact molecular underpinnings of synaptic FUS-mediated regulation of RNA stability and local translation is now an extremely important next step.”*

- *Line 421-422 / 441-422: It would be beneficial to the reader if a brief summary of the behavioural dysfunctions and associated timelines are provided in this manuscript at earlier points. At present, limited information is provided on lines 465-467 and 478-480 after earlier references to the co-submitted manuscript.*

We thank the reviewer for this suggestion, which we agree would be helpful for the reader. We have now added the following information in lines 556-559 of our revised manuscript. *“In particular, $Fus^{\Delta NLS/+}$ mice display increased locomotor activity at 4 months of age, while motor symptoms start at 10 months of age. Moreover, they exhibit impairments in long-term memory consolidation and social inhibition starting from 10 months of age (Scekic-Zahirovic, Sanjuan-Ruiz et al., co-submitted manuscript).”*

Line 477: This sentence needs further elaboration.

We adjusted the discussion to include our latest results from the stability assay and the new results in the co-submitted manuscript. The sentence in question was removed.

- *Line 560. “REF” ?*

Thanks for pointing out this error. We now have added the reference in the lines 690-691 of the revised manuscript as *“Hippocampal cells were then plated onto poly-D-lysine coated 18x18 mm coverslips (Carl ZeissTM - 10474379)”*.

- *Methods cDNA and Quantitative Real-time PCR and Fig. 2c: The Methods section says “Total RNA was...” but in the figure we also see a SNS sample. How was the SNS sample prepared? Were those the same samples/brains used for sequencing and/or imaging? In the figure, what does each dot represent? Was any statistical test performed on those qRT-PCR results?*

We apologize to the referee that we missed to mention the sample description used for cDNA and Quantitative Real-time PCR in the method section and the statistics in Figure 2c. We have used both synaptoneurosomes (SNS) and corresponding total lysate (Total) for cDNA preparation and qPCR. This information has been included in lines 871-872 in the revised manuscript as *“RNA was isolated from synaptoneurosomes (as described before) and paired total cortical lysate from three independent 1-month-old C57/Bl6 mice using QIAzol reagent”*.

These were not the same samples used for imaging or sequencing. However, we used the same method described in method section - line 756 “Preparation of synaptoneurosomes from mouse brain tissues”, for preparing SNS samples for qPCR, CLIP and RNA sequencing, proteomics, electron microscopy and western blot experiments. In figure 2c, each dot represents total and

SNS samples prepared from three different mice. This has been updated in the lines 1367-1369 in the revised manuscript. “ *Bar graphs represent mean \pm SD. Each dot represents total and SNS samples prepared from three different mice. n=3, **p<0.01, unpaired t-test*”.

- *Line 635: The antibody code should be included*

Thank you for this suggestion. We have now added the antibody code in line 780 of our revised manuscript (Santa Cruz, sc-47711).

- *Line 640-641: The methods imply 15 cycles of HiSeq 2500 sequencing were performed for the CLIP-seq libraries. As this would lead to reads that are typically too short to map, can the details be checked and updated accordingly.*

We have included additional information on this point, as requested by the referee in lines 785-788 of our revised manuscript. This is the added text: “*Strand-specific paired-end CLIP libraries were prepared by using The SMARTer Stranded Total RNA-Seq Kit - Pico Input Mammalian (Clontech Laboratories, Inc., A Takara Bio Company, California, USA) without the ribosomal depletion step followed by PCR amplification for 15 cycles. The libraries were sequenced on HiSeq 2500.*”

- *Fig. 3c legend: please, specify error bars and statistics.*

We have added the requested information in the legend of **Figure 3c** (lines 1387-1389) of our revised manuscript: “*Error bars represent mean \pm SD. Each dot represents samples prepared from six different mice. n=6, **p<0.01, unpaired t-test.*”

- *Supplementary Fig. 3b legend: Please, specify error bars and statistics*

We have added the requested information in the legend of our new **Supplementary Figure 4b** (lines 1517-1519) of our revised manuscript: “*Error bars represent mean \pm SD. Each dot represents samples prepared from five different mice. n=5, ***p<0.001, unpaired t-test.*”

- *Line 1269 (Supp fig 4e legend): “significantly increased RNAs...” but in the figure it says downregulated.*

We apologize for this error and thank the referee for noticing it. We have now corrected this error in the legend of our new **Supplementary Figure 5e** (lines 1558-1559) of our revised manuscript. “*Gene ontology (GO) terms enriched among the significantly decreased RNAs at 6 months of age*”.

- *Supp fig 4f legend: mentions samples at 6-months but does not refer to 1-month.*

We thank the referee for noticing it. We have corrected this in the legend of our new **Supplementary Figure 5f** (lines 1561-1562) of our revised manuscript. “**(f) Heatmap from the set of up- and downregulated genes between total cortex samples from $Fus^{\Delta NLS/+}$ and $Fus^{+/+}$ at 1 and 6 months of age**”.

Reviewer #3:

There is still a need for a stronger or novel therapeutical approaches for ALS and FTD. Therefore, uncovering novel roles or broadening current understanding of function of disease relevant proteins, such as FUS, is of utmost importance and is of high interest for the field. In this paper the authors first localize FUS to synapses using super resolution microscopy and then using CLIP-Seq on synaptoneurosomal fractions, determine the RNA interactome of synaptic FUS. They show preference of binding to mRNAs associated with synapse organization and plasticity. Using a mouse model of FUS (characterization of which is contained in the co-submitted manuscript Scekcic-Zahirovic et al.) they show age related misregulation of the GABAergic network. The paper is clear, structured and well written, the methodology is appropriate for the conclusions reached.

Of importance, most of the major findings that are discussed in the discussion section point to observations in Scekcic-Zahirovic et al. co-submitted manuscript, which from the behavioral perspective seem to parallel the molecular changes observed in this paper. Therefore, the fate of this manuscript seems to be very much bound to the Scekcic-Zahirovic et al. co-submitted manuscript and the full value of the publication will depend on joint acceptance.

We thank the referee for the positive assessment of our work.

Comments:

- L130-138 *The study is focused on FUS related changes in the synapses of the hippocampus, so the authors should present more caution in drawing parallels with the submitted study, whose behavioral outcomes point to brain regions associated with FTD.*

We thank the referee for pointing this out. We focused our imaging in the hippocampus as its organization is much simpler compared to the cortical synapses. However, previous studies showed strong similarities between cortical and hippocampal synapses (see citations 1-4 below). Therefore, we think that our observations at the hippocampal synapses are still relevant for the organization of cortical synapses.

1. Somogyi, P., Tamás, G., Lujan, R. & Buhl, E. H. Salient features of synaptic organisation in the cerebral cortex. in *Brain Research Reviews* (1998). doi:10.1016/S0165-0173(97)00061-1
2. Zhu, F. et al. Architecture of the Mouse Brain Synaptome. *Neuron* (2018). doi:10.1016/j.neuron.2018.07.007

3. Contreras, A., Hines, D. J. & Hines, R. M. Molecular specialization of GABAergic synapses on the soma and axon in cortical and hippocampal circuit function and dysfunction. *Frontiers in Molecular Neuroscience* (2019). doi:10.3389/fnmol.2019.00154
4. Distler et al. Proteomic Analysis of Brain Region and Sex-Specific Synaptic Protein Expression in the Adult Mouse Brain. *Cells* 9, 313 (2020).

- L177. What percentage of FUS colocalizes with spinophilin? How close is this colocalization? Just by eyeball it seems quite significant. Figure 1e should also have an overlay and quantification for spinophilin.

We did not perform quantification of this marker as it is quite difficult to image a full spine in 2D using dSTORM imaging. Our observation did not suggest an enrichment of FUS in the spine compared to the presynaptic element.

- L182. What is the comparison of synapsin 1 with spinophilin?

We did not quantify the distance between spinophilin and FUS, therefore there is no statistical comparison between synapsin 1 and spinophilin. Spinophilin is a PP1 binding protein used to label the entire spine (Allen et al, PNAS 1997; <https://pubmed.ncbi.nlm.nih.gov/9275233/>). For more precise quantification, we decided to use PSD-95 and Glutamate receptors instead.

- Fig1h is slightly at odds with SupFig1d Imaris image, which shows a lot more postsynaptic FUS signal than what is presented in the F1h (only two spots).

We are sorry about this confusing point. Actually, as detailed in the next point below, the schematic shown in **Supplementary Figure 1d** depicts the imaging and analysis process and does not contain any specific biological events presented in this paper. We have clarified this in the legend now as explained below. The schematic in **Figure 1h** is the one summarizing our findings. Based on the close proximity distance calculation, we found that more than 60% of the FUS signal was in closer proximity to Synapsin 1 (presynaptic marker) compared to any post synaptic markers.

- SupFig 1d and h. It would be clearer if the colors in the scheme had a 'mini-legend' next to the scheme. Though one can follow the green FUS, the number of different labelings makes other marker proteins harder to follow.

The schematic shown in **Supplementary Figure 1d** depicts the imaging and analysis process. The colors chosen are random and do not reflect any specific biological events presented in this paper. This schematic is only shown for illustration purposes, while the schematic in **Figure 1** represents the summary of our results and has a legend. To avoid confusion, we added this clarification in the legend of supplementary figure 1 in lines 1479-1481 of our revised manuscript: "Colors and structures depicted in this panel do not represent specific labels or structures, and are only shown for illustration purposes."

- *SuplFig2a. figure needs correcting. 'Post' is written across one of the blots.*

We thank the referee for noticing this error. We have corrected this figure.

- *L273. Superfluous sentence*

We agree that this sentence was not necessary and we have now deleted it in our revised manuscript.

- *L404. Considering their importance for neurodegeneration the authors should comment on the changes observed for APP (and APLP1) as is shown in supl fig5.*

We thank the referee for this suggestion. In the discussion, lines 630-635 of our revised manuscript, we have now added the following sentences: *"It hasn't escaped our attention that among the altered synaptic FUS targets is App, encoding the amyloid precursor protein (APP), a transmembrane protein with crucial roles in synaptic function and stability⁸⁶⁻⁸⁸. Importantly, APP is causally linked with Alzheimer's disease and our observations suggest a previously unidentified link between synaptic FUS accumulation and APP misregulation. The recent observation that APP contributes to the regulation of inhibitory synapses^{89,90} reinforces this link."*

- *L629 Add a reference for the CLIP-seq method.*

We thank the referee for noticing this. We have now included the references in the CLIP-method section in the revised manuscript in line 774. *"We performed this protocol following a previously published method^{22,53}."*

REVIEWER COMMENTS

Reviewer #1 (Remarks to the Author):

The authors have made several additions to the manuscript which have significantly strengthened the work. The new assessment of RNA stability through ActD/RNA-seq experiments are particularly helpful for defining the consequences of synaptic FUS accumulation and a mechanism for these consequences. I recommend publishing, with only two minor concerns:

- On lines 261-262, the authors state "The exclusively nuclear non-coding RNA Malat1 (Metastasis Associated Lung Adenocarcinoma Transcript 1) was strongly detected in total cortex samples..." Please provide a reference for this statement, since in some conditions Malat1 is not exclusively cytoplasmic — see, for example, PMID 30150986.

- The new experiments involving the effect of FUSdNLS on RNA stability are excellent. How do the results of these experiments compare with those of Kapelli et al. (PMID 27378374), who similarly examined RNA stability by treatment of neural precursors with ActD upon FUS knockdown?

Reviewer #2 (Remarks to the Author):

The authors have made considerable effort to address the comments of both myself and the other reviewers. On the whole I am satisfied with the changes, particularly the analysis of RNA stability that provide a new connection through the paper, and believe the manuscript has been strengthened. I have only minor concerns remaining.

I still have the concern that differential expression calling with $\log_{2}FC > 0$ is challenging with this sample number and read depth. Whilst some justification is provided by the rebuttal, the size of change is minimal and not supported by any independent methods. A qPCR of key examples being followed would be merited.

Line 128: Should read "thereby mimicking the majority of ALS-causing FUS mutations."

Figure 5c: x-axis label should be made clearer – currently it appears as the title of the plot, and part of the x-axis labels are covered by the key. The same is true of the related figure in the supplementary.

Line 298: "a critical role" seems too presumptive based on the evidence presented thus far. The terminology should be revised so as to not overstate.

Line 454: Should "Most RNAs with increased accumulation" read "Most of these 28 FUS-interacting RNAs with increased accumulation". If not, it implies that all 485 increased synaptic RNAs are bound by FUS in the following section

The summary of the co-submitted manuscript is again suggested to be earlier in the text (i.e. in introduction or results). The latest additions are again very late in the text.

The manuscript may benefit from general shortening to aid readability.

Reviewer #3 (Remarks to the Author):

These authors have adequately addressed my concerns raised in the previous version. I have one additional suggestion. In general FUS binds to RNA quite indiscriminately mRNA, ncRNA, introns, etc. As the authors have prepared and sequenced a poly A library for sequencing I suggest that for increase clarity throughout the manuscript and figures the authors use 'mRNA' instead of just general 'RNA'. Barring this small additional note, I recommend this revised manuscript for publication. I would like to congratulate these authors on an important manuscript in the field of ALS, which I feel will have significant impact.

Reviewer #1 (Remarks to the Author):

The authors have made several additions to the manuscript which have significantly strengthened the work. The new assessment of RNA stability through ActD/RNA-seq experiments are particularly helpful for defining the consequences of synaptic FUS accumulation and a mechanism for these consequences. I recommend publishing, with only two minor concerns:

We thank the referee for the positive assessment of our revised manuscript and for recommending publication.

- On lines 261-262, the authors state "The exclusively nuclear non-coding RNA Malat1 (Metastasis Associated Lung Adenocarcinoma Transcript 1) was strongly detected in total cortex samples..." Please provide a reference for this statement, since in some conditions Malat1 is not exclusively cytoplasmic — see, for example, PMID 30150986.

We thank the referee for pointing this out. We have added the relevant citations and updated our manuscript to accurately describe Malat1 localization in lines 255-257 as follows: "The nuclear-enriched non-coding RNA Malat1 (Metastasis Associated Lung Adenocarcinoma Transcript 1) was strongly detected in total cortex samples, with only trace coverage seen in synaptoneurosomes (**Supplementary Fig. 3b**)."

- The new experiments involving the effect of FUSdNLS on RNA stability are excellent. How do the results of these experiments compare with those of Kapelli et al. (PMID 27378374), who similarly examined RNA stability by treatment of neural precursors with ActD upon FUS knockdown?

The study by Kapelli et al. tested the role of FUS in RNA stability in human neural precursors, but not in mouse mature neurons. To clarify this point in our manuscript and put these data in context of published work, we added the following sentences in lines 474-477 of our manuscript: "While it has been shown before that knock down of FUS in primary cortical neurons⁷⁴ and human neuronal precursor cells⁶⁰ leads to altered stability of mRNAs, the effect of FUS accumulation on mRNA stabilization has never been addressed before."

Reviewer #2 (Remarks to the Author):

The authors have made considerable effort to address the comments of both myself and the other reviewers. On the whole I am satisfied with the changes, particularly the analysis of RNA stability that provide a new connection through the paper, and believe the manuscript has been strengthened. I have only minor concerns remaining.

We are glad that the referee finds that our manuscript was strengthened by the revisions and we also found the referees' comments constructive and helpful in our revision process. We hope that this updated version will clarify the remaining minor concerns of this referee.

I still have the concern that differential expression calling with $\logFC > 0$ is challenging with this sample number and read depth. Whilst some justification is provided by the rebuttal, the size of change is minimal and not supported by any independent methods. A qPCR of key examples being followed would be merited.

We understand the referee's concern and agree that the changes, even though strongly significant, are actually small, a finding that we think is biologically relevant. However, we do not agree that qPCR quantification is the best method for validation of our RNA-seq results, since it is a more biased and noisy method than our deeply sequenced RNA-seq experiment with 6 replicates per condition and at least 30 million reads per sample. It is also widely recognized that RNA-seq measurements are reproducible across platforms and the overall correlation between RNA-seq and qPCR is very high (e.g. Figure 5a in reference PMID: [25150838](https://pubmed.ncbi.nlm.nih.gov/25150838/)). Thus, we do not expect that qPCR would result in more accurate measurements of the logFCs than RNA-seq.

Moreover, we do not have any material left from the initial samples and we would need at least six months to generate new mice to repeat the 6-month time point of this sequencing. We strongly believe that delaying publication of this work for this validation would not be justified.

However, we can show that the results are indeed reproducible, via an independent sequencing experiment of these samples. This was done in a different facility, namely a Next Generation Sequencing company in Denmark (<https://omiics.com>), with different library preparation, i.e. riboZero depletion, and a different analysis pipeline for estimating differential expression, i.e. DESeq2 in R. We computed the log2FC between FUSdNLS and WT of 6-month-old SNS and compared the values to the log2FC of FUS targets reported in Figure 4c in our manuscript. As shown in the figure below, we observe a strong correlation of the logFC between the two experiments, confirming the reported logFCs. We hope that this analysis convinces this referee of the reproducibility of our findings.

Line 128: Should read “thereby mimicking the majority of ALS-causing FUS mutations.”

Thanks for pointing out this error. The sentence in the line 127-128 of the revised manuscript has been changed to “thereby mimicking the majority of ALS-causing FUS mutations.”

Figure 5c: x-axis label should be made clearer – currently it appears as the title of the plot, and part of the x-axis labels are covered by the key. The same is true of the related figure in the supplementary.

We thank the referee for pointing out this mistake. X-axis labels have been now corrected in Figure 5c and Supp. fig 9b as suggested, in the revised manuscript.

Line 298: “a critical role” seems too presumptive based on the evidence presented thus far. The terminology should be revised so as to not overstate.

We thank the referee for making this suggestion. We have now changed the sentence in line 298 of our revised manuscript as follows: ‘*Our data suggests that FUS may play an important role in maintaining synaptic integrity and organization.*’

Line 454: Should “Most RNAs with increased accumulation” read ““Most of these 28 FUS-interacting RNAs with increased accumulation”. If not, it implies that all 485 increased synaptic RNAs are bound by FUS in the following section

We thank the referee for noticing this mistake. We have now changed the sentence in line 461-462 of our current manuscript as follows: “*Most of these 28 FUS-interacting RNAs with increased accumulation*”.

The summary of the co-submitted manuscript is again suggested to be earlier in the text (i.e. in introduction or results). The latest additions are again very late in the text.

We agree with the referee that the summary of the co-submitted manuscript is very late in the text. We have corrected this and the summary has been added in the result section in line 408 of the revised manuscript as follows “*These early synaptic changes mechanistically explain the behavioral dysfunctions that these mice develop (Scekic-Zahirovic, Sanjuan-Ruiz et al., co-submitted manuscript). In particular, $Fus^{ANLS/+}$ mice display increased locomotor activity at 4 months of age, while motor symptoms start at 10 months of age⁴⁴. Moreover, they exhibit impairments in long-term memory consolidation and social inhibition starting.*”

The manuscript may benefit from general shortening to aid readability.

We agree with the referee on this point and have shortened our manuscript, also to adhere to the editorial recommendations of Nature Communications.

Reviewer #3 (Remarks to the Author):

These authors have adequately addressed my concerns raised in the previous version.

We thank the referee for the constructive feedback and for the positive assessment of our revised manuscript. We appreciate the time and effort put in this evaluation.

I have one additional suggestion. In general FUS binds to RNA quite indiscriminately mRNA, ncRNA, introns, etc. As the authors have prepared and sequenced a poly A library for sequencing I suggest that for increase clarity throughout the manuscript and figures the authors use 'mRNA' instead of just general 'RNA'. Barring this small additional note, I recommend this revised manuscript for publication. I would like to congratulate these authors on an important manuscript in the field of ALS, which I feel will have significant impact.

We have followed the referee's suggestion and specified all places in text and figures, in which we are unambiguously referring to mRNA for clarity. In addition to the abstract and text, this was changed in figures and legends listed below:

- Fig 2c
- Fig 4b
- Fig 4 legend
- Fig 5e
- Fig 5 legend
- Supplementary Fig4 legend
- Supplementary Fig 5 legend
- Supplementary Fig 9 e
- Supplementary Fig 9 legend

REVIEWER COMMENTS

Reviewer #2 (Remarks to the Author):

I am satisfied with the latest round of revisions and author responses. I am happy to recommend for publication.

REVIEWERS' COMMENTS

Reviewer #2 (Remarks to the Author):

I am satisfied with the latest round of revisions and author responses. I am happy to recommend for publication.

We thank the referee for evaluating our last round of revisions and for recommending our manuscript for publication